



# Modeling the integrated framework of water resources system considering socioeconomic development, ecological protection, and food production: A practical tool for sustainable water uses

Yaogeng Tan[1], Zengchuan Dong[1], Xinkui Wang[1], Wei Yan[2]

1. College of Hydrology and Water Resources, Hohai University, Nanjing 210098, China

2. School of Geographic Sciences, Xinyang Normal University, Xinyang, 464000, China

Correspondence to Zengchuan Dong (zcdong@hhu.edu.cn)

**Abstract:** The accelerated consumption of water resources caused by the rapid increase of population and urbanization is intensifying the negative interactions of water uses across socioeconomic development, ecological

protection, and food productivity (SEF), which causes an imbalance between water supply & demand, ecological and food vulnerability, and further water unsustainability. To solve this problem, this study developed an integrated modeling framework to better identify the dynamic interaction and coevolution process of the nexus across SEF systems in the context of sustainable water uses by coupling two models: system dynamic model (SD) and multi-objective optimization model. First, the SD model is used to simulate both the dynamic interaction of each agent and

the coevolution process of the whole system under external changes. Next, the multi-objective optimal model is used to generate the optimal scheme by iteration process with the initial scheme of SD, further identifying the dynamic interaction and coevolution process in terms of sustainable water use. Finally, the model uncertainty considering different weighting factors is analyzed. The framework is applied to the Upper Reaches of Guijiang River Basin, China. Results show that: (i) the rapid economic growth intensifies the ecological awareness and cannot support such

rapid development because it rises the conflict between environment and economic water uses, resulting in more water shortages of socio-economy and food agent. (ii) Once the economic growth rate decreased, water resources are able to support economic development with decreased overload index and stable crop yield, which further contributes to water sustainability. (iii) The river ecological agent is the critical factor that affects the robustness of the model. (iv) The equal consideration of each water usage is the most beneficial to sustainable development. These results

highlight the importance of water resources management considering the tradeoffs across multiple stakeholders and give a strong reference to policymakers for comprehensive urban planning.

**Key words:** Sustainable water uses; SD model; optimal model; coevolution process; feedback linkages

## 1. Introduction

In recent years, the rapid increase of economic development and urbanization is accelerating the consumption

of water resources, further contributing to the imbalances and conflicts between water supply and demand (Carpenter et al., 2011; Yaeger et al., 2014; Perrone and Hornberger, 2014). The accelerated consumption of water resources not only influences the natural hydrological cycle and furthermore rainfall process, affecting the agricultural water uses and eventually giving the vulnerability of food safety, but also reduces the ecological streamflow, deteriorating the river ecological health and affecting the aquatic biodiversity (Bei et al., 2009; Yang et al., 2019; Tan et al., 2019). The

resulting huge pressure on food safety, river ecosystem, and socioeconomic development presents the characteristics of universality and complexity, seriously restricting the achievement of regional sustainable development goals (Walter et al., 2012; Liu et al., 2014; Yang et al., 2019). Therefore, detecting the sustainable balance across the



different water needs has become one of the hotspots of water resources planning and management communities (Baron et al., 2002; Falkenmark, 2003; Rockstrom et al., 2009; Perrone and Hornberger., 2016; Zhang et al., 2018; Luo and Zuo, 2019). At present, water resources system is composed of numerous water sectors and is susceptible to the influence of external conditions, intensifying their complex dynamic interactions under external changes (Phillips, 2001; Thomas, 2001; Liu et al., 2007a; Parker et al., 2008; Wagener et al., 2010; Secchi et al., 2011; Yaeger et al., 2014). The dynamic interactions are usually characterized by high dimensionality and non-linearity, which challenges the goal of sustainable water uses (Gastélum et al., 2010; Yaeger et al., 2014). Thus, identifying the coevolution process and dynamic interactions across multiple water uses is one of the crucial and effective approaches on how the water resources system performs more sustainably (Sivapalan et al., 2012; Collins et al., 2011; Yaeger et al., 2014; Thompson et al., 2013; Wagener et al., 2010).

There are lots of approaches that tackle water resources problems facing multiple water users, such as optimal algorithms (Abdulbaki et al., 2017), decision support system (Chandramouli and Deka, 2005), Multi-criteria Decision Analysis (MCDA) (Afify, 2010), etc., which can be classified as systematic analysis approach (SAA). SAA is one of the most effective methods to solve the water resources management problems characterized by their complexity and was carried out by many scholars (Faridah et al., 2014; Liu et al., 2008; Li et al., 2015; Jia et al., 2015). The complexity of the water resource system is usually embodied by its different spatial subsystems that are reflected in terms of physical relations between upstream and downstream. In addition, each subsystem includes multiple water sectors, which contributes to its characteristics of large-scale and high-dimensionality. Among all the SAA, system optimization approach is one of the most practical approaches to manage complex water resources systems in a nonlinear, integrated, and comprehensive way (Moraes et al., 2010; Singh, 2014; Chen et al., 2017; Li et al., 2019a). It gives insights on how to allocate the water resources on a regional or watershed scale in a balanced way (Li et al., 2015; Liu et al., 2019). However, optimal approach usually puts emphasis on how to attain the optimal value of each individual water user and neglects the dynamic interactions and relations among these users.

The core content of sustainable water resources is to emphasize the value of water resources and the protection of the ecological environment while ensuring socioeconomic development and food security (Gohari et al., 2013). It stresses the relevance and dynamic interactions of those water uses instead of their individual properties. In this respect, the term "nexus" is emerged to reveal the multiple components and their interlinkages within a system. This term is first conceived by World Economic Forum (2011) to promote and discuss the indivisible relationships between the multiple uses of resources. It provides the universal rights of water, energy, and food, and developed the water-energy-food (WEF) nexus framework (Hoff, 2011; Biggs et al., 2015). However, nexus thinking includes but not limited to WEF (Duan et al., 2019), such as water-energy-food-environment nexus (Hellegers et al., 2008), energy-water-environment (EWE) nexus (Shahzad et al., 2017), water-power-environment (WPE) nexus (Feng et al., 2016, 2019), etc. In addition, the components of the water resources system also include the interaction between the natural hydrological cycle and human society, which can be regarded as a human-natural nexus system and is usually assessed on a watershed scale (Liu et al., 2007b). Although those nexus systems are made of different components, their common feature is that the coevolution and feedback process of such components are considered in a dynamic and integrated way.

In the latest years, many new technical methods based on systematics are emerged to deal with the problem of performances and interactions of a complex system in a more advanced and comprehensive way. Nair et al., (2014) stressed the energy uses in an urban water system are from both water supply and wastewater, and suggested that life cycle analysis (LCA) is the widely used approach in the water-energy nexus. LCA is addressed based on the different



stages of the evolution of the whole system and its components. Apart from LCA, Ecological network analysis (ENA)
is another systematic method that can provide a consolidated analysis for both direct and indirect flows reflected in
complicated chains of production and consumption, indicating the potential to investigate the trade-off between
multiple elements (Chen and Chen, 2016). System dynamic (SD), that based on the computer simulation method, is
one of the most visualized approaches for analyzing information feedback systems (Forrester et al., 1971). It can link
different elements for analyzing the dynamic simulation under different external conditions. Its ability to dynamically
simulating the system characterized by non-linearity, multiple feedbacks, and complexity makes it popular among
many scholars (Venkatesan et al., 2011; Li et al., 2018; Yang et al., 2019). Although those advanced systematic
methods made decent contributions on simulating and characterizing a real system, there are still some shortcomings
and limitations in applying to comprehensive water resources management: (1) those methods are used to simulate
the dynamic status and feedbacks just in an objective way but no optimal function inherently, which limits the goal
of sustainable water uses to some extent; (2) optimization algorithms are commonly addressed on water resources
planning and allocation facing multiple water users, but rarely evaluated in a dynamic way and usually failed to assess
their dynamic interactions. Therefore, an integrated framework of water resources management by coupling both
systematic methods and optimal approaches needs to be further studied.

The objectives of this paper are, therefore, (1) to develop an integrated modeling framework that couples the
water uses across the socioeconomic development, ecological protection, and food production (SEF) in a complex
system perspective and explore its dynamics under external changes by using system dynamic model, (2) to apply
the framework in the upper reaches of Guijiang River Basin (UGRB) by coupling the SD model and multi-objective
model, in order to explore their dynamic interaction and coevolution process in an optimal way to achieve sustainable
water use and (3) identify the model uncertainty to assess the various tradeoffs to stakeholders and recognize the
main factor(s) that most influences the model robustness to improve the reliability of the integrated framework. In
doing so, we are able to identify both the coevolution process and dynamic interactions of complex water resources
systems and how to achieve sustainable water uses considering multiple sectors for water resources management
communities.

## 2. Methodology

### 2.1 Outlines of the integrated modeling framework

Sustainable water uses is composed of that for socioeconomic development (i.e., **Socioeconomic**), ecological
protection (i.e., **Ecology**), and food security **(**i.e., **Food)** and their interactions (Hunt et al., 2018; Uen et al., 2018;
Perrone and Hornberger., 2016; Feng et al., 2019), which is investigated as SEF nexus. The external changes that
affect the performances and interactions of water resources systems can be addressed by the "pendulum model"
outlined by Kandasamy et al. (2014). He stressed that the term "pendulum swing" refers to the shift in the balance of
water utilization between economic development and environmental protection. It has periodic changes that can be
classified into several stages in a relatively long-term period. In short, it can be classified into the "initial" stage that
productivity is about to emerge, "developing" stage that production activities are negatively affecting the environment,
and "environmental protection" stage to which environmental issue is paid great attention. The detailed description
of the "pendulum model" can be found in **Supplementary material S1**.

The integrated modeling framework of the water resources system comprises two models: system dynamic
model that discloses the dynamic interaction process of the whole system; and the multi-objective optimization model



that generates the optimal water allocation scheme that considers the needs of each agent. The overall research framework of the integrated system considering two models is shown in **Fig.1**, and the detailed model description is

provided in the following sections. First, the external drivers of the whole nexus system are the changes in the development level of socio-economy that can be separated into several time steps (here we use "τ" to nominate). Both the initial ecological streamflow and the initial water supply scheme, along with their interactions can be simulated by SD model under each τ. Next, the initial scheme is acted as the input of the optimal model (Li et al., 2018), and the optimization result is generated by iteration of the optimal algorithm with the initial value. The iteration process

will not be terminated until the adjacent iteration result is within the specific error. Then, the optimization result will transfer back to update the system status of the current τ, and start a new simulation with the next τ. If τ=T, end the whole process, otherwise, repeat this process. Here T is the total length of simulation time. Finally, the dynamic process of the water resources system can be embodied by the trajectories of system variables connecting each τ, including water supply/demand, carrying capacity, ecological flow, food production (or crop yield), etc.

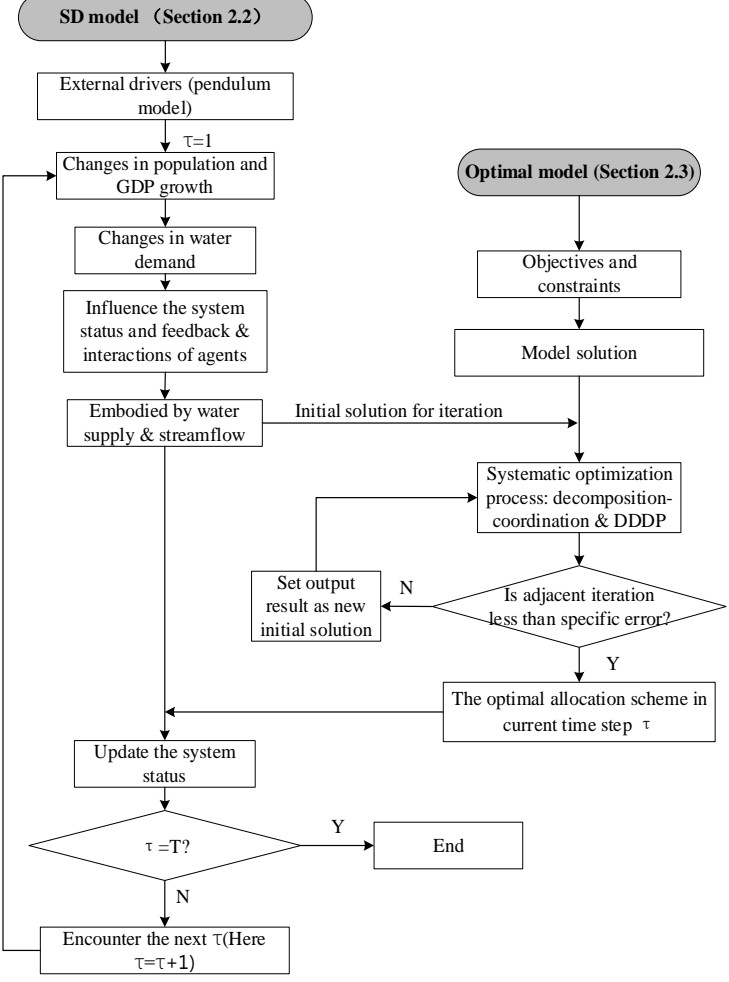


**Fig.1** Overall research framework of the integrated modeling approach



### 2.2 SEF system developed by system dynamic model

#### 2.2.1 Socioeconomic agent

The socioeconomic agent describes the regional population rate, urbanization rate, and GDP products. Their
dynamic changing process in water recipient regions can be described within the logistic model, which can be
expressed by the following differential equations (Jørgensen and Bendoricchio, 2001; Feng et al., 2019):

$$\frac{dN}{dt} = rN, \frac{dI}{dt} = rI \tag{1}$$

where N and I are population size and the total amount of GDP, r is the natural growth rate of GDP or population.
The natural growth rate can be assessed by collecting and analyzing the statistical data of urban population, rural
population, and the total amount of GDP (including primary, secondary, and tertiary industry). The water demand of
socioeconomic agent can be outlined by the following equation:

$$WD_{dom} = \frac{q_{dom} \times N \times d}{1000} \tag{2}$$

$$WD_{indus} = I_{GDP} \times q_{indus} \tag{3}$$

where $WD_{dom}$ and $WD_{indus}$ are the annual domestic (including urban and rural) and industrial (including secondary
and tertiary) water demand ($m^3$), $q_{dom}$ and $q_{indus}$ are the domestic and industrial water usage quota, which means daily
water consumption per person (L/person/day) and water consumption of the industrial added value per $10^4$ Yuan
($m^3/10^4$ Yuan), respectively. It should be also noted that economy also includes agricultural economy. For agricultural
economy, the economic basis of farmer's response is reflected by average incomes that can be expressed by the
following:

$$I = \frac{1000\sum_{i=1}^{n} Y_i p_i}{N_r} \tag{4}$$


where I is the farmer's average income, $Y_i$ is the ith crop yield, $N_r$ is the rural population. Crop yield is a significant
component of both primary industry value and can measure farmers' income because farmers sell these foods to
customers and get profits. The calculation of crop yield is shown in Section 2.2.3. The system dynamic model of the
Socioeconomic agent is presented in Fig.2. The external changes outlined by the pendulum model are exactly
embodied by the changing rate of population and GDP expressed by Eq.(1). In the scope of SD, the dynamic process
of population growth can be expressed as follow:

$$population(\tau) = population(\tau - d\tau) + (net\ population\ growth) \times d\tau \tag{5}$$

The dynamic growth of the three industries is similar to Eq.(5). From Fig.2 we can see that the changing population
and GDP (i.e., the external drivers), will result in the changing water demand, which further affects the water supply
and eventually the status of the entire system (See 2.2.4).



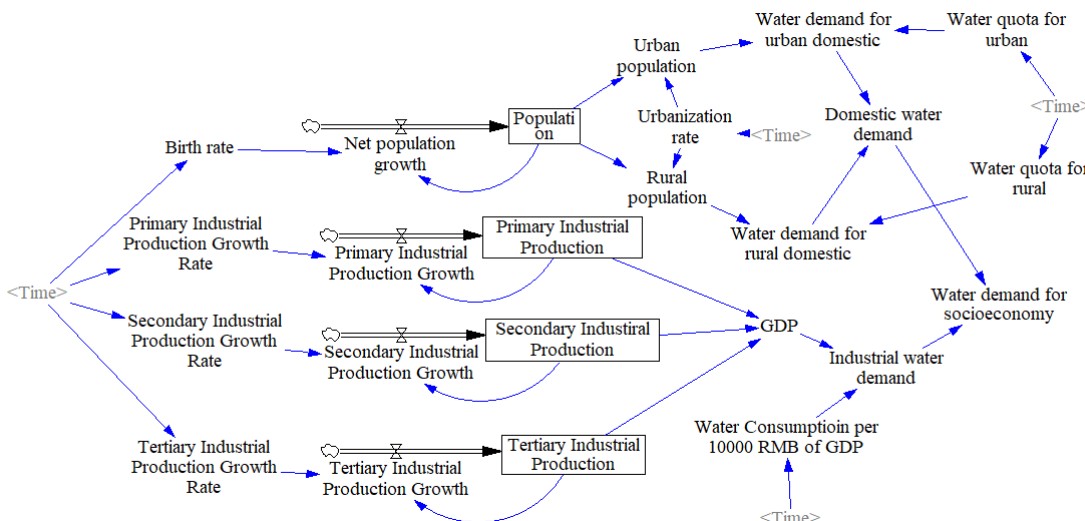

Fig.2 The inner stimulus and feedbacks of socioeconomic agent

### 2.2.2 Ecological agent

Ecological water demand includes vegetation and river streamflow. Ecological water demand of vegetation is used to maintain the physiological function of canopies. The method of evaluating the amount of vegetation ecological demand is based on their evapotranspiration that can be treated as the water gap (Shi et al., 2016; Saxton et al., 1986):

$$WD_{veg} = K_s \cdot K_c \cdot ET_0 - P_e \qquad (6a)$$

$$ET_0 = \frac{0.408\Delta\left(H_{net}-G\right)+\gamma\dfrac{900}{T+273}u_2\left(e_0-e_z\right)}{\Delta+\gamma\left(1+0.34u_2\right)} \qquad (6b)$$

$$K_s = \frac{\ln\left[100\times\dfrac{S-S_w}{S_c-S_w}+1\right]}{\ln 101} \qquad (6c)$$

where $WD_{veg}$ is the vegetation water demand. $P_e$ is the effective precipitation. $ET_0$ is potential evapotranspiration based on the Penman-Monteith equation, and the particular variables can be seen in Neitsch et al., (2011). $K_s$ and $K_c$ are soil moisture and canopy coefficients, respectively, which denotes the ratio of maximum water demand and potential evapotranspiration. $S$, $S_c$, and $S_w$ are the coefficient of actual, wilting, and critical soil moisture, respectively. For river streamflow, the Tennant method is adopted in this study:

$$W_{eco} = 86400\times\sum_{m=1}^{12}d_m Q_m P_m \qquad (7)$$





where $W_{eco}$ is the ecological streamflow in the annual average level ($m^3$), $d_m$ is the day number of month m, $Q_m$ is the observed streamflow ($m^3/s$). $P_m$ is the percentage of observed streamflow of the month m. It should be noted that the river streamflow calculated by Eq.(7) is just the initial value with given $P_m$'s, and it will be input to the optimal model for optimized solution.

### 2.2.3 Food agent

The food agent is mostly related to agricultural water usage, including crop water requirements based on phenological stages. It is also the fundamental condition of primary industry and farmer's incomes (See 2.1.1). For crops, water usage is related to crop yield. The main water supply is provided by irrigation. We use the crop coefficient method to estimate crop water demand based on the Food and Agricultural Organization report No. 56 (FAO-56) (Allen et al., 1998). For each crop, its growth process can be separated into several stages that have different potential crop water demands (Allen et al., 1998; Smilovic et al., 2016):

$$W_p = \int_{t_0}^{t_n} K_c(t) \cdot ET_0 dt \tag{8a}$$

$$W_a = W_p - P_e \tag{8b}$$

where $W_P$ is potential crop water demand, and can also be called reference crop demand of crop i, $K_c(t)$ is the crop coefficient of stage t for a specific crop, $t_0$ and $t_n$ is the first and last stage of the growth process of a specific crop. $W_a$ is the irrigation water demand. The maximum crop yield is based on the hypothesis that the crop water supply (including precipitation) can meet $W_p$ (Allen et al., 1998). According to FAO-56, crop growth is usually divided into four phenological stages: initial, development, middle, and end, and corresponds to three different crop coefficients: $K_{c,ini}$, $K_{c,mid}$ and $K_{c,end}$. For details, see Allen et al. (1998). For each crop, the crop yield is presented as follow (Smilovic et al., 2016):

$$\frac{Y_s}{Y_p} = \prod_{t=t_0}^{t_n} \frac{Y_{s,t}}{Y_{p,t}} = \prod_{t=t_0}^{t_n} \left[ 1 - K_{y,t} \left( 1 - \frac{W_{s,t} + P_{e,t}}{W_{p,t}} \right) \right] \tag{9}$$

where $W_{s,t}$ is the actual irrigation water supply for crop i at time t, $Y_s$ and $Y_p$ is the crop yield under actual and ideal condition (both irrigation water supply $W_s$ and precipitation $P_e$ can meet the crop water demand $W_p$), $K_{y,t}$ is yield response factor of the crop i at time t. Due to the limitation of local water resource conditions, crop water supply is usually equal to or less than crop water demand. That is, $(W_s + P_e) \leq W_p$, and crop water supply is greatly related to crop yield. The value of $Y_s/Y_p$ is also equal to or less than one, and it takes the "=" sign when the crop yield attains the maximum. In this case, the water supply also attains the maximum.

It should be noted that the agricultural and vegetation water demand in the future is hard to predict accurately compared with because they are related to meteorological and land use variables, which cannot be predicted precisely on a long-time scale. Fortunately, the statistical characteristics of regional weather data are usually assumed to be consistent on a multiyear scale (Feng et al., 2019). That is, the characteristics of the future precipitation can be capture by multiannual historical data. Therefore, the average level of water demand of historical multi-year is proposed in this study because historical data can represent the hydrological conditions of a certain area.

### 2.2.4 Overall simulation of SD and update of the system status

The overall simulation of the SD model is to reveal the dynamic interactions influenced by dynamic external drivers. The dynamic interactions are embodied by the feedback linkages/loops among different agents (includes their





water supply and demand). The update process of the SD model is reflected by some relevant variables that are greatly affected by water supply of different agents. The relevant variables include water shortage (aiming at all
agents), carrying capacity & overload index & farmer's income (socioeconomic agent), the deviation between ecological and observed streamflow (ecological agent), and crop production (food agent) (See Table 1). Fig.3 outlined the update process and interaction of each agent of SD model and how the initial water supply and other variables are simulated. The feedback linkages among socioeconomic and ecological agent under external drivers are revealed by follows:

●    Population (+) → Domestic water demand (+) → Domestic water supply (+) → Ecological streamflow (-)

     ●    GDP (+) → Industrial water demand (+) → Industrial water supply (+) → Ecological streamflow (-)

     ●    Domestic/Industrial water supply (+) → Carrying Capacity (+) → Population/GDP (+)

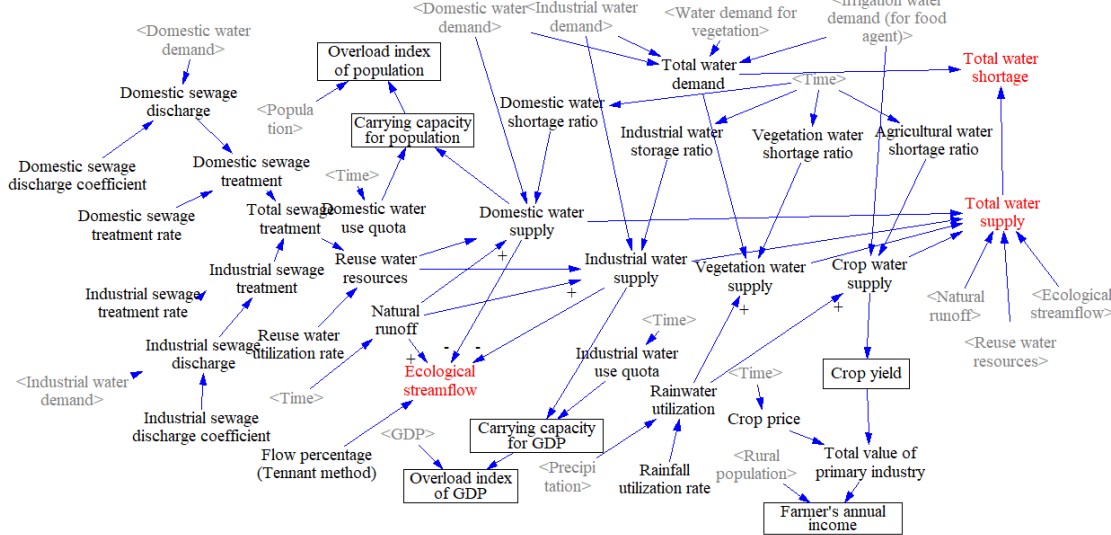

Fig.3    Total simulation and update process of SD model

Here "carrying capacity" quantifies the population/GDP that can be supported by a certain amount of water resources. The overload index is given by dividing predicted population/GDP by carried population/GDP. The higher value of the overload index, the more serious degree of overload. The feedback linkages also occur in other agents. For example, socioeconomic agent affects food agent and finally transfers back to socioeconomic itself:

     ●    Precipitation (+) → Crop/vegetation water supply (+) → Crop yield (+)

●    Population (+) → Food demand (+) → Crop water supply (+) → Crop yield (+)

     ●    Crop yield (+) → Crop carrying capacity (+) → Population (+)

     ●    Crop yield (+) → Primary industrial production (+) → Farmer's income (+) → GDP (+)

     Here "crop carrying capacity" quantifies the population size that can be supported by a certain amount of crop yield. Those feedback linkages can be expressed by the causal loop diagram (Fig.4). The symbol "+" and "-" beside
the arrow is the positive/negative feedback linkage, respectively. The clockwise arrow with a "+" inside is a positive feedback loop, while the counterclockwise arrow with a "-" inside is a negative feedback loop.



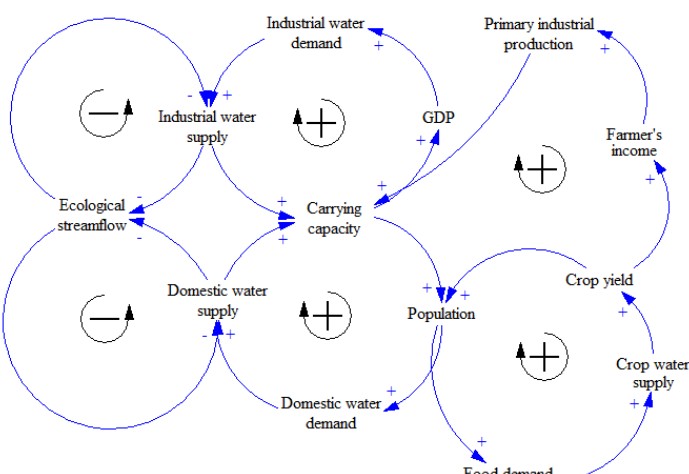

Fig.4 Causal loop diagram

Adequate water supply is one of the most important conditions to ensure socioeconomic development and is
also a prerequisite for crop production. Therefore, in socioeconomic agent, policymakers expect to decrease the water
shortage by increasing water supply to ensure socioeconomic development, since increased population/GDP is
accompanied by increased water demand (Li et al., 2019b). Then, the increasing water supply leads to the increased
carrying capacity, and the population/GDP will increase again. Such linkage can be regarded as a positive feedback
loop. Similarly, in food agent, increased population intensifies food demand, and more water supply is needed to
increase the crop yield, which can eventually support more population size. This linkage can also be regarded as a
positive feedback loop. Adequate water supply can be embodied by the following equation by minimizing the water
shortage ratio:

$$WS_j = WD_j \times \left(1 - WSR_j\right) \tag{10}$$

where $WS_j$, $WD_j$, and $WSR_j$ are water supply, water demand, and water shortage ratio for jth sector, respectively.
Here j is each component of the SEF system. Crop and vegetation water supply also include effective precipitation
($P_e$).

It should be noted that water supply expressed by Eq.(10) is just the expected value for policymakers. However,
water for socioeconomic development and river ecological health always conflicts with each other as both of them
consume natural runoff. In the scope of SD, it is embodied by the negative feedback loop. That is, the increased
(domestic and industrial) water supply will contribute to decreased river streamflow that deteriorates the ecological
health (Yin et al., 2010; 2011; Yu et al., 2017), and vice versa. To consider this issue, a certain percentage of
streamflow (usually for ensuring basic flow) are the rigid constraint for the ecological agent, and the water supply
considering ecological basic flow is expressed as follow:

$$WS = \min(\sum_{j=1}^{J} WS_j, R + W_{reuse} - W_{eco}) \tag{11}$$

where R and $W_{reuse}$ is the natural runoff and reused water (includes rainfall utilization and recycled). The water supply





presented in Eq.(11) is the initial water supply simulated by SD.

However, it is still not enough for considering each aspect of water use. If the adequate water supply is ready for ensuring socioeconomic development and crop yield, the ecological streamflow will be decreased. Even the ecological basic flow is ensured, the ecological function of a river will be limited. Therefore, the optimization model
is presented in this study to achieve the sustainable water uses of each agent (see next section) by inputting the initial simulated result of SD and iteration (Li et al., 2018). The simulated result is calculated by Eq.(11). Finally, the optimal scheme of water supply and ecological streamflow is transferred back to SD model to update the status of the current time step, and then all the dynamic changes of each variable can be assessed. Other variables and equations can be seen in **Supplementary materials S2**.

270                                         Table 1   Main equations for model update

| Variables | Units | Mathematics | Remarks |
|---|---|---|---|
| Water supply for each sector | $10^8 m^3$ | Corresponding water demand × (1 − corresponding water shortage ratio) | This is valid for each sector. For example, if calculating domestic water supply, just multiply domestic water demand with domestic water supply coefficient. Others are the same. |
| Total water supply | $10^8 m^3$ | min (Water storage + reuse water resources – ecological streamflow, sum (Domestic water supply, Industrial water supply, Agricultural water supply, Vegetation water supply)) | See Eq. (11) |
| Ecological streamflow | $10^8 m^3$ | Water storage × Flow percentage | Tennant method, see Eq.(7). The initial percentage is set as 0.2 (Oct~Mar) and 0.4 (Apr~Sep) to consider the basic streamflow |
| Carrying capacity: population | people | Domestic water supply × 1000/(water quota for domestic × day of a certain year) | The unit of water quota for domestic is L/people/d. Both urban and rural are calculated like this. |
| Carrying capacity: GDP | $10^8$ yuan | (Industrial water supply + tertiary water supply) / Water consumption per 10000RMB of non-agricultural industry + Total value of primary industry | |
| Crop yield | $10^4 t$ | Crop yield is the nonlinear function of crop water supply and demand, see Eq.(9) | |
| Overload index | | Predicted economic index/ Carrying capacity | Valid for both population and GDP |
| Total value of primary | $10^8$ yuan | Crop production × crop price per unit | |



| | | | |
|---|---|---|---|
| industry | | | |
| Farmer's annual income | $10^4$ yuan | Total value of primary industry/rural population | Eq.(4) |

*2.3 Optimization approach of the integrated system*

*2.3.1 Model conceptualization*

In a water system inside a watershed or a region, there are multiple water supply projects to different water users. This system in a watershed is called a "large water resources system" (**Fig.5**a). It is subdivided into multiple sub-watershed or subregions that are called "subsystems" (**Fig.5**b). In this case, reservoirs can provide not only socio-economic developments but also environmental impacts. They are constructed across the rivers for both water supply of the whole region or watershed and adjust the downstream river streamflow, which should be considered individually to target the river ecology concerns.

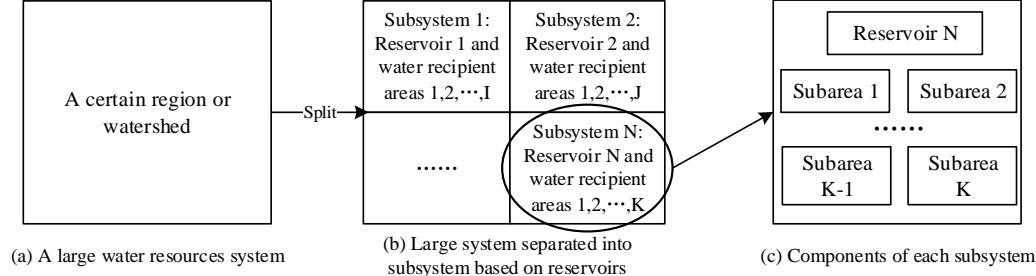

(a) A large water resources system  (b) Large system separated into subsystem based on reservoirs  (c) Components of each subsystem

**Fig.5**  Water resources system and its decomposition

The whole system is separated into subsystems that contain one individual reservoir and several corresponding water recipient areas (**Fig.5**b) as there is usually more than one reservoir in a certain region. We call these subsystems "reservoir supply subsystem". Such a subsystem can be further separated into the smallest unit: a reservoir and each water recipient region (or called "subarea") (**Fig.5**c). In this view, the total system of the water resources in a certain region (watershed) can be divided into several subsystems or subareas that consist of a three-level hierarchical structure.

It should be noted that the term "large water resources system" is not the same thing as the framework of SEF system presented in this study. To combine these two terms, each agent of SEF nexus system can be distributed to each subarea (with the objective of food, socio-economy, and vegetation) and reservoir (river ecology) (see **Fig.6**). Therefore, we can coordinate these objectives to achieve sustainable development by setting up multi-objective optimal model.

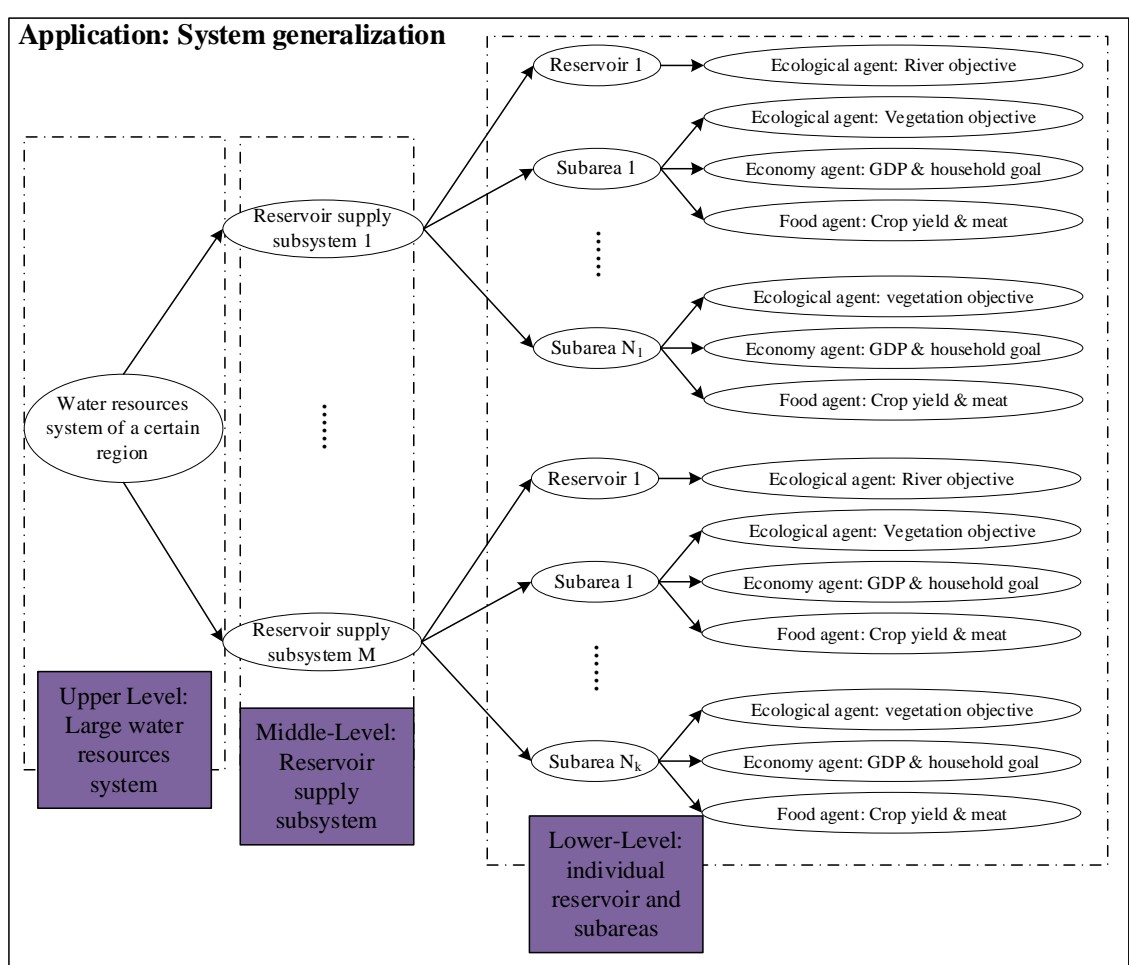

**Fig.6** Large water resources system considering the SEF nexus

*2.3.2 Objective function*

(1) Agent of socioeconomic development

The objective of socioeconomic agent is expressed by the minimum water shortage rate:

$$F_{socemy} = \frac{1}{T} \min \sum_{k=1}^{K} \sum_{t=1}^{T} \left( \frac{WD_{socemy,kt} - WS_{socemy,kt}}{WD_{socemy,kt}} \right)^2 \tag{12}$$

where $F_{socemy}$ is the objective function of the socioeconomic agent. WD and WS are the total water demand and supply (including reservoir and other water projects) of this agent. T is the time length of the reservoir operation horizon. Subscript k and t are the number of subarea and time steps, respectively. It should be noted that farmer's income affiliated in the socioeconomic agent is greatly related to crop yield. Thus, this goal will be discussed in food agent.

(2) Agent of ecological protection

Ecological protection comprises two aspects: river ecology and vegetation ecology. For river ecology, the





artificial intervention in the natural flow regime is a crucial factor of the severe deterioration of river ecosystems
(Shiau et al., 2013; Tan et al., 2019). It has been proved that the term "amended annual proportional flow deviation"
(AAPFD) is used to embody the river's health degree and used in many studies in term of river ecology, and assumed
that the minimum deviation between observed (natural) and actual streamflow contributes to the healthy status of
river ecological health (Gehrke et al., 1995; Ladson and White, 1999; Liu et al., 2019; Feng et al., 2019). The objective

function can be expressed as follow:

$$F_{riv} = \min \frac{AAPFD}{5} = \min \frac{1}{5n} \sum_{j=1}^{n} \sqrt{\sum_{m}^{12} \left( \frac{W_{eco,mj} - QN_{mj}}{QN_j} \right)^2} \tag{13a}$$

where the subscript "riv" represents river ecology, QN is the observed streamflow. The variable AAPFD ranges from
zero to five and the minimum value represents the best status of river's ecological health (Gehrke et al., 1995; Ladson
and White, 1999; Yin et al., 2010). Thus, we divided it by five to normalize the objective function and make it range

from zero to one. The subscript n, m, and j are the total year number, mth month, and jth year.

Vegetation, similar to the river environment, is also an indispensable part of ecology because it produces oxygen
to improve air pollutions and purifies water bodies. The abundant water supply contributes to these goals. Therefore,
the objection of vegetation is expressed as follow:

$$F_{veg} = \frac{1}{T} \min \sum_{k=1}^{K} \sum_{t=1}^{T} \left( \frac{WD_{veg,kt} - WS_{veg,kt}}{WD_{veg,kt}} \right)^2 \tag{13b}$$

where the subscript "veg" represents vegetation ecology.

The objective of the ecological agent is reflected by maintaining both aspects, reflected by the following
normalized form (from zero to one):

$$F_{eclgy} = \frac{F_{veg} + F_{riv}}{2} \tag{13c}$$

where $F_{eclgy}$ is the total objective function of the ecological agent.

(3) Food agent

The goal of the food agent is to maximize crop yield and is the indispensable condition of increase primary
industry products and farmer's income. Also, food is the most fundamental prerequisites for people's survival and
farmer's income. The mathematical expression is presented as follow:

$$F_{food} = \max \sum_{n=1}^{N} \left( \frac{Y_a}{Y_p} \right)_n \tag{14a}$$

where N and L are the total number of crops and livestock, respectively. $Y_a$ and $Y_p$ are the crop yield under the actual
and ideal conditions, respectively.

The calculation of food yield is based on the Food and Agricultural Organization report No. 56 (FAO-56) (Allen
et al., 1998). According to the crop yield equation based on FAO-56 (see Eq.(9)), crop production that determines
farmer's profit is directly related to irrigation water (FAO, 2012; Liu et al., 2002; Lyu et al., 2020). Therefore, the

maximum supply of crops (includes both precipitation and artificial water supply for crops) is the most critical
condition for maximum crop yield. Thus, the normalized objective of the food agent can be rewritten as:





$$F_{food} = \frac{1}{T} \min \sum_{k=1}^{K} \sum_{t=1}^{T} \left( \frac{WD_{food,kt} - WS_{food,kt}}{WD_{food,kt}} \right)^2 \qquad (14c)$$

### 2.3.3 Tradeoffs between objectives

As can be seen in objective functions, three benefits are set minimum (Eqs.(12)(13c)(14c)), which may
contribute to the conflict between objectives. The tradeoffs across SEF nexus can be reflected by Pareto frontier that
can describe a set of non-dominated optimal solutions that any one of these three objectives are unable to be improved
unless sacrificing other objectives (Reddy and Kumar, 2007; Feng et al., 2019; Beh et al., 2015; Burke and Kendall.,
2014). We can reclassify all the water users from each of the three agents into two categories: Instream and off-stream
water users (Hong et al., 2016). River ecological water demand can be regarded as an instream water user, and all
others can be considered as off-stream water users. Therefore, according to the objective function expressed by
Eqs.(12), (13c), and (14c)), the weighted objective function can be rewritten by:

$$\min F = F_{socemy} + F_{eclgy} + F_{food} = \alpha \left( F_{socemy} + F_{veg} + F_{food} \right) + \theta F_{riv}$$

$$= \sum_{j=1}^{J} \sum_{k=1}^{K} \sum_{t=1}^{T} \alpha_j \left( \frac{WD_{jkt} - WS_{jkt}}{WD_{jkt}} \right)^2 + \theta \frac{1}{5n} \sum_{j=1}^{n} \sqrt{\sum_{m}^{12} \left( \frac{Q_{mj} - QN_{mj}}{\overline{QN_j}} \right)^2} \qquad (15)$$

where ($F_{socemy}$+$F_{veg}$+$F_{food}$) is off-stream water users, and $F_{riv}$ is the instream water users. The subscript j is the index
of the off-stream water users, respectively. j=1,2,3 represents socio-economic, food, and vegetation water usage,

which corresponds to the subscript "socemy", "eclgy" and "food". α and θ are weight factors and $\sum_{j=1}^{J} \alpha_j + \theta = 1$.

Previous literature demonstrated the optimal solution shaped like Eq.(15) is Pareto-optimal because of the positive
weights and concave objectives, and the non-dominated sorting process is used to find the optimal solution of Eq.(15)
because the characteristic of either concave or convex is difficult to be proven (Marler and Arora., 2009; Feng et al.,
2019; Goicoechea et al., 1982; Zadeh, 1963). For each given combination set of α and θ, the optimal solution can be
attained by decomposition-coordination (DC) principle and discrete differential dynamic programming (DDDP) (see
section 2.3.5).

The tradeoff across objectives is reflected in the values of multiple sets of weighting factors
$\boldsymbol{r} = \left( \alpha_1, \alpha_2, \alpha_3, \theta \right)^T$, revealing different decision-makers' preferences. Considering that the contradictions also
occur in off-stream water users, the balanced priority should be addressed to consider each off-stream water user
(Casadei et al., 2016), that is, $\alpha_1$=$\alpha_2$=$\alpha_3$. Therefore, the tradeoff and decision preference between instream and off-
stream is reflected by the values of θ (0≤θ≤1). The larger value of θ represents more concerns about river
ecology. In this study, the parameter θ is initially set as 0.5 to give an equal consideration of both instream and off-
stream water usage. It should be noted that this weight combination is one possible set that considers the equal use
of instream and off-stream water uses, and different weight of weighting factor reveals the preferences of stakeholders.
Different vectors of $\boldsymbol{r}$ can affect the performance of SEF nexus and are used to assess the uncertainty and robustness
of the model to improve its reliability (see Section 5.2 & 5.3).

### 2.3.4 Constraints

The model constraints include the connection of subsystems, the water balance equation, and the upper and





lower limits. The details are found in **Supplementary material S3**.

*2.3.5 model solution*

The SEF model of water resources sustainability is a compound system that is classified into multiple hieratical structures (**Fig.6**). Therefore, the model solution of this structure should be solved by systematical analysis techniques. In this study, we use the decomposition-coordination (DC) method to solve this sophisticated model. The core procedure of this method comprises two parts: firstly, the large system is decomposed by several subsystems (i.e.,

reservoir and recipients) using Lagrange function considering the interrelations between subsystems, and its coordination process is performed by coordination variables; secondly, the optimization process using DDDP method of each subsystem. The monthly historical streamflow observations with the length of decades are the important model input for DDDP method (i.e., subscript t in the variables of the entire optimal model), assuming that the characteristics of future streamflow are captured by the historical data (Feng et al., 2019). The detailed descriptions

are found in Supplementary material S4. The entire procedure for the overall framework of the model is outlined below:

Step1: Initialize the parameters, including initial reservoir storage, water recession coefficient, the total amount of water resources of the recipient area, etc.

Step2: For each reservoir supply system, calculate the initial water supply of each subarea and reservoir

streamflow at $\tau=1$ (set as $S_0$). These variables can be simulated by SD model (see 2.2.4and Table 1).

Step3: Using DDDP algorithm to optimize each subsystem decomposed by Lagrange function with coordinate variables. The expression of coordinate variables is the function of the initial scheme, which is shown in **Supplementary material S4**. To use DDDP, the width of the corridor is given (set as $\Delta I$), and the traditional DP is optimized within $\Delta I$. Mark the result generated by DP (include both water supply and river streamflow) as S1. If $|S_1-$

$S_0|<\varepsilon$, go to the next step, otherwise repeat this step.

Step4: Narrow the width of the corridor and continue the DP process, set Si as the optimal result, where i is the iteration number. If $|S_i-S_{i-1}|<\varepsilon$, go to the next step, otherwise repeat this step.

Step5: Update the coordinate variables and compare them with the initial coordinate variables. If the error is within $\varepsilon$, the optimal solution (i.e., water supply and streamflow) will be generated, otherwise, repeat step3~5.

Step6: Optimize the next reservoir supply subsystem by repeating step2~5, and the summary of each subsystem is the global optimal solution.

Step7: The optimal result in Step6 is under $\tau=1$, and prepare to encounter the next time step ($\tau=2$) of external drivers by repeating overall procedures until $\tau=T$.

*2.4 Sustainable development degree (SDD) assessment*

The SEF nexus is a complex system with all ecological, socioeconomy, and food systems, or agents as we called in this study, affecting water resources. A proper SEF balance provides resource support to achieve sustainable development. Therefore, the three agents should be considered to evaluate the sustainable development degree. We selected the indicators listed in **Table 2** based on the three agents and are used to evaluate the impact of sustainable development.

**Table 2** Sustainable development evaluation index system of three agents

| Agent | Indicators | Property |
|---|---|---|
| Socioeconomy | Overload index of population | - |
|  | Overload index of GDP | - |





| | Per capita GDP (RMB/people) | + |
|---|---|---|
| | Water consumption per 10000RMB of GDP ($m^3/10^4$RMB) | - |
| | Farmer's income (RMB/people) | + |
| Food (Agriculture) | Crop production (t) | + |
| | Effective irrigation area for crops ($km^2$) | + |
| Ecology | Effective irrigation area for vegetation ($km^2$) | + |
| | AAPFD | - |

The property (+, -) of indicators denotes positive and negative indicators, respectively. The positive/negative indicators mean they have positive (negative) impacts on the corresponding agent and were termed as a development/constraint index (Yang et al., 2019). Considering the ranges of indicators listed in **Table 2** are different, they should be normalized before evaluation. The positive and negative indicators normalization is shown by Eq.(16a) and (16b).

$$y_{ij} = \frac{x_{ij} - \min\limits_{i=1}^{m} x_{ij}}{\max\limits_{i=1}^{m} x_{ij} - \min\limits_{i=1}^{m} x_{ij}} \tag{16a}$$

$$y_{ij} = \frac{\max\limits_{i=1}^{m} x_{ij} - x_{ij}}{\max\limits_{i=1}^{m} x_{ij} - \min\limits_{i=1}^{m} x_{ij}} \tag{16b}$$

where $x_{ij}$ and $y_{ij}$ is the original and normalized indicator j in sample i, and m is the total number of samples. The entropy weight method is then adopted to calculate SDD, which calculates the information entropy of indicators that reflect their relative change degree on the whole system (Wang et al., 2019). The information entropy of indicator j in sample i is expressed by:

$$E_j = -\frac{1}{\ln m} \sum_{i=1}^{m} d_{ij} \ln d_{ij} \tag{17a}$$

$$d_{ij} = \frac{y_{ij}}{\sum\limits_{i=1}^{m} y_{ij}} \tag{17b}$$

Finally, the entropy weight of each indicator is expressed by:

$$\omega_j = \frac{1 - E_j}{\sum\limits_{j=1}^{n} (1 - E_j)} \tag{18}$$

where n is the total number of indicators in a certain agent.

The SDD is calculated based on the coupling coordination degree (Sun and Cui, 2018), reflecting the degree of coordination of various factors or subsystems. In this study, SDD is calculated based on the coordination of three agents (SEF) and expressed by:





$$SDD = \sqrt{C_1 C_2} \tag{19a}$$

$$C_1 = \left[ \frac{SOCEMY(t) \cdot ECLGY(t) \cdot FOOD(t)}{\left( SOCEMY(t) + ECLGY(t) + FOOD(t) \right)^3} \right]^{\frac{1}{3}}$$

$$= \left[ \frac{\sum\limits_{p=1}^{P} \omega_{pj} y_{pj} \cdot \sum\limits_{q=1}^{Q} \omega_{qj} y_{qj} \cdot \sum\limits_{r=1}^{R} \omega_{rj} y_{rj}}{\left( \sum\limits_{p=1}^{P} \omega_{pj} y_{pj} + \sum\limits_{q=1}^{Q} \omega_{qj} y_{qj} + \sum\limits_{r=1}^{R} \omega_{rj} y_{rj} \right)^3} \right]^{\frac{1}{3}} \tag{19b}$$

$$C_2 = \frac{1}{3} \left( SOCEMY(t) + ECLGY(t) + FOOD(t) \right)$$

$$= \frac{1}{3} \left( \sum\limits_{p=1}^{P} \omega_{pj} y_{pj} + \sum\limits_{q=1}^{Q} \omega_{qj} y_{qj} + \sum\limits_{r=1}^{R} \omega_{rj} y_{rj} \right) \tag{19c}$$

where SOCEMY(t), ECLGY(t), and FOOD(t) are the coordination degree of socioeconomy, ecology, and food agent, respectively. P, Q, R is the total indicator number in socioeconomy, ecology, and food agent.

**3. Study area and data sources**

*3.1 A brief description of the study area*

Guijiang River Basin (GRB) is one of the most imperative branch basins of the Pearl River Basin (PRB) in South China. PRB belongs to the typical karst area and is the second-largest river basin in China in terms of total runoff and also the third largest river basin in terms of total area. The upper reach of Guijiang River Basin (UGRB)
(24°6' ~25°55'N, 110°~111°20'E) is selected as a case study as it represents the highly conflicts between socio-economic growth and ecological protection in karst areas. Furthermore, reservoirs are widely constructed in UGRB to supply water for socio-economy but are likely to deteriorate the river ecological health by alternating natural flow (Yin et al., 2010; 2011). UGRB is also a karst area with a total area of 13,131 km$^2$, with about three million people. Also, UGRB has a total crop planting area of about 2,400 km$^2$, a total vegetation area of about 3,700 km$^2$, and yearly
average precipitation of about 1600mm. UGRB is located in Guilin City and refers to eight administrative regions (or counties). Seven reservoirs are constructed in UGRB to provide water resources support for maintaining the development of socio-economy. The detailed parameters of seven reservoirs and their three-level hieratical structure, including subareas, are found in **Supplementary material S5**. Guilin city is both a heavy industrial city and a national major tourist city, and the population and economic development will keep rapidly increasing in the near future. It
will exacerbate the conflicts between social development, food safety, and environmental protection, especially for water use of river ecological environment, resulting in severe ecological deterioration of the lower Guijiang River basin and even lower XRB. Therefore, how to achieve coordination and sustainable development in UGRB between these aspects is becoming a challenging problem in the upcoming years and is necessary to be solved.

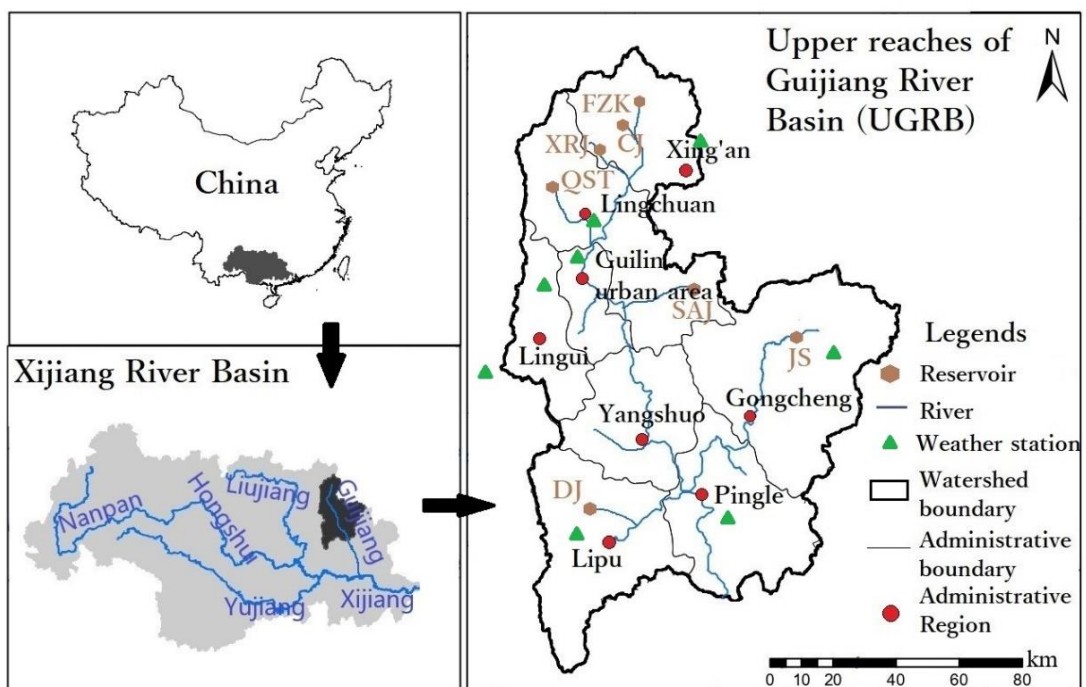

**Fig.7**   A brief location of UGRB

*3.2 Datasets and parameter initialization*

Datasets of the case study include socio-economic, water use, land use, meteorological and hydrological data. The major source of socio-economic data, including population and GDP, are the statistical yearbooks of both Guilin City and Guangxi autonomous region from 2005-2014. The Municipal Government of Guilin City (MGGC) predicted

population and GDP till 2045, along with per capita water use from the water industry standard of the People's Republic of China, to predict the water demand of socioeconomic agent (Venkatesan et al., 2011). These predicted socioeconomic indexes are exactly the external drivers of the whole integrated modeling framework (see Section 2), and the corresponding growth rate in different stages are shown in **Table 3**. Water use data include historical water usage and total water amount found in Guilin water resources bulletin (2005~2014). Land use data contain the spatial

distribution of crops and vegetations with a resolution of 1km×1km that can be found in the Resource and Environment Data Cloud Platform, China Academy of Sciences (REDCP-CAS). The crops in the study areas are mainly corns, rice, and vegetables, and their crop coefficients are found in FAO-56 (the detailed values are found in **supplementary material S6**). Meteorological data from 1956 to 2013, including daily average wind speed, sunshine duration, maximum and minimum temperature, relative humidity, and precipitation, are found in meteorological

stations. The hydrological data from 1958 to 2013, including the monthly inflow of each reservoir, can be found in hydrological stations for the input of optimal algorithm. All the initialized parameters and the total index of the data sources can be found in **Supplementary material S6**

**Table 3**   External drivers (i.e. socio-economic changes) of the entire research framework based on pendulum model





| Yearly growth rate (%) | Stage 1 (2021~2025) | Stage 2 (2026~2035) | Stage 3 (2036~2045) |
|---|---|---|---|
| Population | 1.23 | 3.41 | 1.24 |
| Secondary industry | 1.99 | 4.11 | 2.36 |
| Tertiary industry | 3.04 | 5.33 | 1.24 |

## 4. Results

*4.1 Coevolution process of SEF nexus*

The coevolution trajectories of population, GDP, water supply & demand, streamflow, and objective function ($F_{SOCEMY}$, $F_{eclgy}$, $F_{food}$, based on Eq.(10)(11)(12)) referring to each component of the SEF nexus is shown in **Fig.8**. As can be seen in **Fig.8**, the coevolution process of all the items depicts the characteristics of different stages. Finally, the (quasi-)stable state is converged, i.e., the variations of each variable are small or close to zero. It happens because

the rate of external changes in the last stage (i.e., economic indexes) is much lower than in the previous stage, which decreases the internal changes (i.e., Streamflow water and three objective functions). In the first stage, the growth rate is relatively low and is based on the historical data, and the growth rate of $F_{socemy}$, $F_{eclgy}$, and $F_{food}$ is also slow. When entering the second stage, the economic growth will lead to increased water demand. However, according to the achievement of sustainable development based on the optimal model, ecological concerns should not be neglected.

Therefore, the increase of river streamflow will also happen driven by the optimal model to maintain the river ecological health, consequently reducing the total water supply and increasing the water shortage of water users (**Fig.8**c). As $F_{food}$ and $F_{SOCEMY}$ can reflect the water shortage of the corresponding water users, their value will also increase sharply (**Fig.8e** and **8g**) due to the rapid increase of socio-economic indexes. When entering the last stage, the development of socio-economy will tend to stable, and the increasing speed of $F_{food}$ and $F_{SOCEMY}$ will decrease

compared with that in the second stage. This is because the relatively stable development of socio-economy does not need too much increased streamflow water (i.e., the increase rate of streamflow water is closed to a relatively stable state), and both changing rates of water supply and demand tend to be stable consequently (**Fig.8**c).

We can also see that the water supply system competes for the instream ecological system. As shown in **Fig.8**, especially in stage 2, increased streamflow is accompanied by increased $F_{SOCEMY}$ and $F_{food}$ (**Fig.8e** and **8g**), reflecting

the decreased satisfaction degree of the water supply of socio-economy and agriculture, thereby revealing the competition use between instream and off-stream water uses. The tradeoff between instream and off-stream water users can be obtained by the optimal model to solve for the best coordination status between them by adjusting economic development modes and balance the priority of each water users. It should be noted that the ecological objective ($F_{eclgy}$) is in a relatively stable status in all stages compared with other objectives (**Fig.8**f). This is because

the ecological agent contains not only river streamflow but also vegetation. The booming economy drives the optimal model to focus more on river ecological health ($F_{riv}$), and there are limited water resources for off-stream water users including vegetation. The dual effect of increasing streamflow water and decreasing water for vegetation makes the $F_{eclgy}$ relatively stable. However, the optimal model takes the effect that the optimal allocation scheme is obtained by shifting streamflow water because instream and off-stream water use is intrinsically conflicted with each other, and

should be coordinated by adjusting different weights of each component (see section 6).

**Fig.8** Coevolution process of SEF nexus model

*4.2 Dynamic interactions of SEF system*

*4.2.1 Socioeconomic-ecology response linkages*

**Fig.9** illustrated the loop of socioeconomy-ecology feedback. As demonstrated in **Fig.9**, the response linkage of





carrying capacity and overload index involves the changes of economic indexes, water supply & demand, and streamflow water (Feng et al., 2019). In the beginning, the economy is still increasing slowly, and the increasing rate of water demand is also slow. The population and GDP are near the carrying capacity in this stage (i.e., the value of OI is near 1). In the following stage, both increasing population and GDP intensify the water demand (**Fig.9a** and **b**).

To satisfy socio-economic development demands, water supply of economic agent has also increased. However, there will be a more significant concern of the river ecological system (**Fig.8**c, **Fig.9**c) because ecological streamflow is an important part of sustainable water use, simulated by the optimal model. In this view, the growing rate of water supply of domestic and industry (**Fig.9**d) will be less than the growth rate of water demand (**Fig.9**b) and therefore contributes to the increase of water shortage, which is in accordance with the performance shown in **Fig.8**e. The

increasing water shortage will generate the gap between carrying capacity (**Fig.9**e) and predicted economic indexes (**Fig.9**a). Then, the overload index will further increase, consequently affecting socio-economic development. It further contributes to the overload of the water resources system, which even restricts the socio-economy instead. In the last stage, as the growth rate of population and GDP alleviates (**Fig.9a**), there will be a relatively slower increase rate of streamflow water, and there will be more water space for socio-economic development. Although the water

shortage is increasing, its rate is lower than that in the second stage. The carrying capacity will be able to catch the predicted economic index if the stable or slower growth rate continues. The overload index is also decreased (**Fig.9f**, and the whole system tends to be stable. This response linkage indicates that the excessive population and GDP growth will eventually lead to increased overload status by increased ecological streamflow, and moderate growth of socioeconomy will promote the best status of both each agent and the entire system, and eventually promote

sustainable water use.



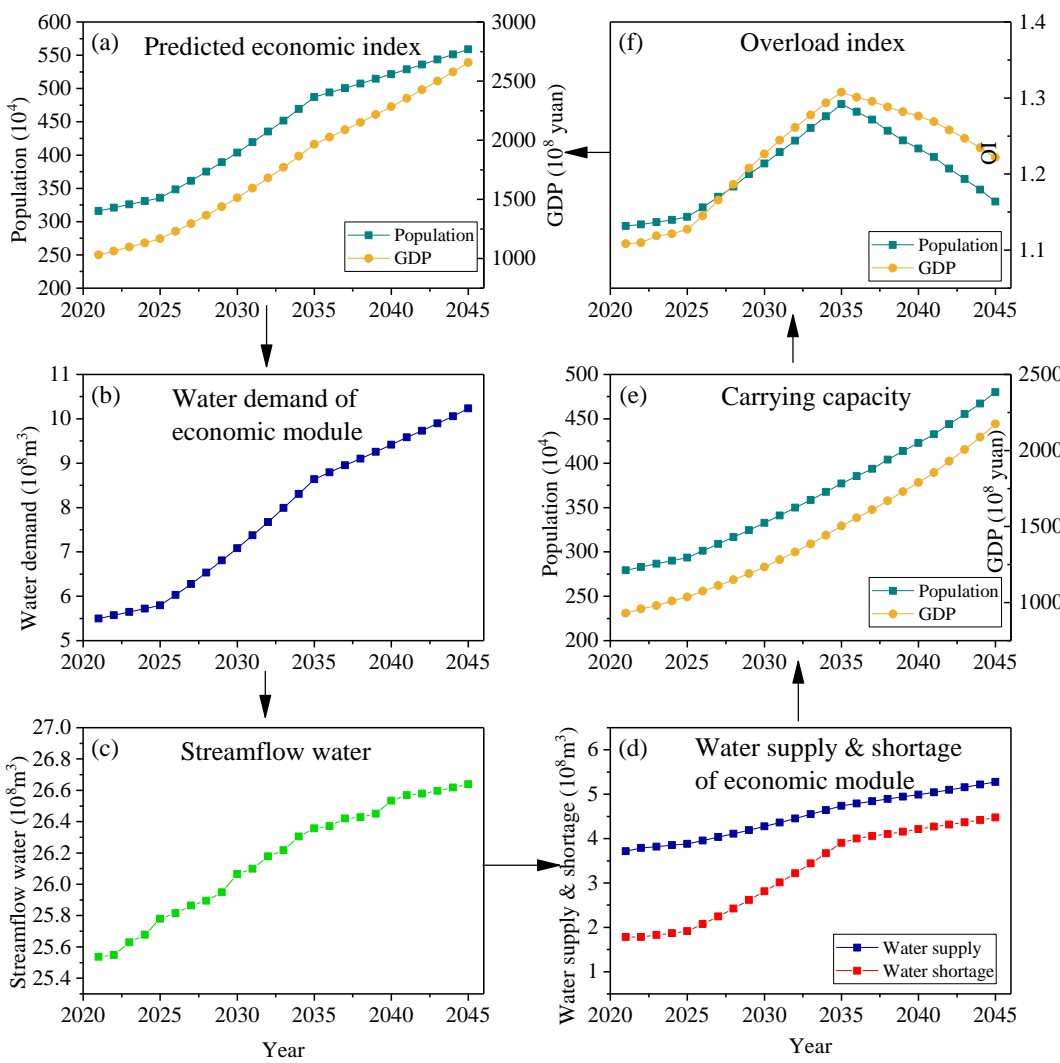

**Fig.9**  Response linkage of socioeconomy-ecology feedback loop

*4.2.2 Ecology-food response linkages*

Another performance is the ecology-food response linkage and is shown in **Fig.10**. It not only illustrated the linkage between food and ecological water usage but also demonstrated the coevolution of ecology components of both instream (river ecology) and off-stream (vegetation) aspects. We can see from **Fig.10** that the increased streamflow water is the driving force of the ecology-food response. However, the increasing streamflow water was driven by the rapidly increasing socio-economic scale. The optimal model is used to achieve the goal of sustainable development to balance the need of different users, especially that of instream and off-stream. The increased streamflow has two effects on the ecology-food response linkage. First, the variable $F_{riv}$ describes the ecological





health of a certain river. According to the definition of AAPFD, the higher value of streamflow water indicates the lower value of $F_{riv}$, which indicates that the river ecology is getting better. Second, the increasing streamflow water restricts the water supply of all off-stream water users, including agricultural and vegetation water (**Fig.10**b). Irrigation and vegetation water use is the largest off-stream water consumer, and their increased water shortage was
also driven by increased streamflow water (**Fig.10**d).

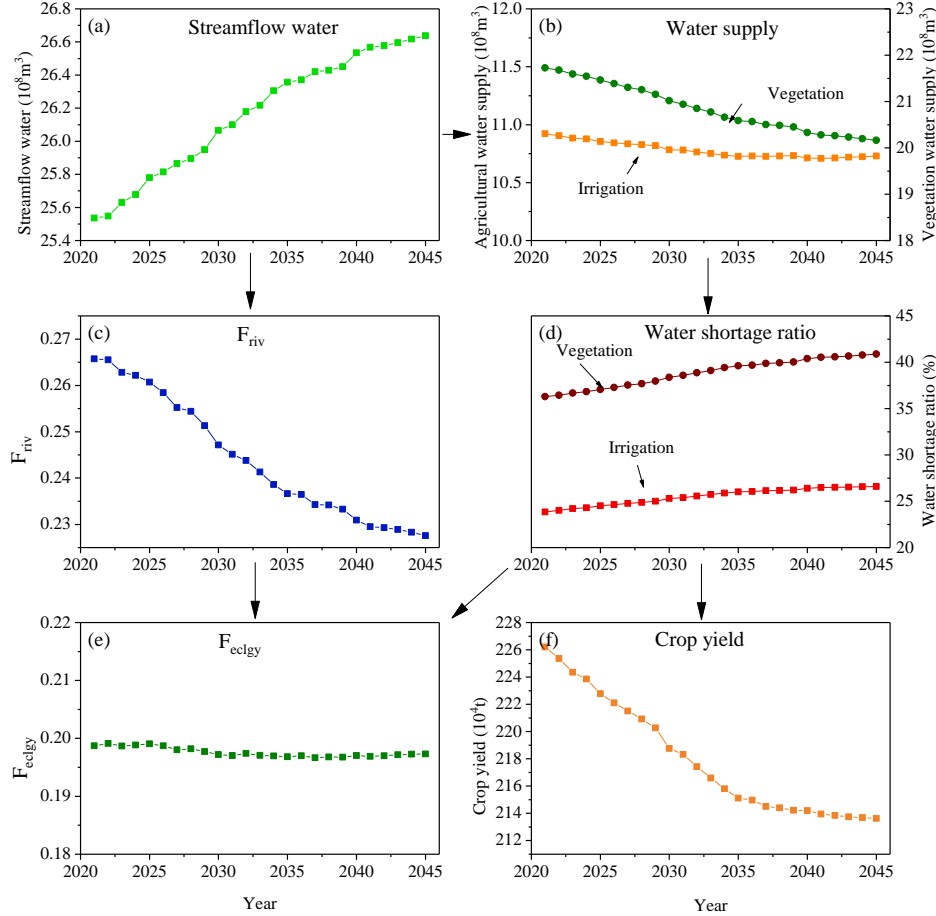

**Fig.10**   Ecology-food response linkage

For ecological agent, the dual effect of increased streamflow water and decreased vegetation water makes the stable change of $F_{eclgy}$ (**Fig.10**e), indicating that the ecological aspect of UGRB is maintaining a good status. For
food agent, crop yield is strongly affected by the satisfaction degree of irrigation water, and the increased water shortage of crop water will, therefore, indicate the decrease of crop yields (**Fig.10**f). But it tends to be stable in stage 3 because of the slower growth rate of socioeconomic index, which contributes to the stable changing trend of streamflow water and further contributes to the stable changes of crop production.





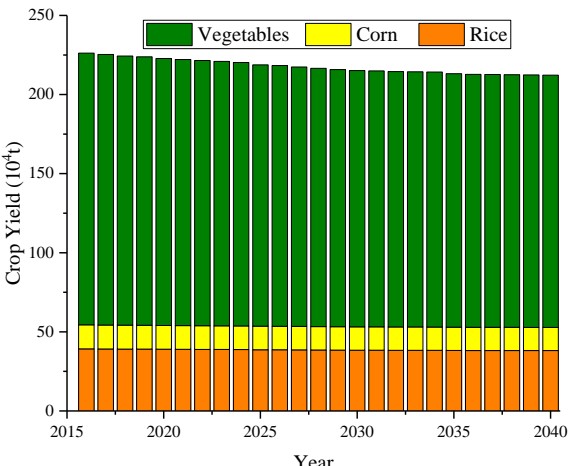

**Fig.11** Detailed crop yield

*4.2.3 Socioeconomy-food response linkage*

Because food safety ensures people's survival, The decreased crop yield is driven by the increased streamflow water that also caused an increasing overload index (**Fig.9**f) in the second stage, which is reflected by socioeconomy-food response linkage (**Fig.12**). The detailed crop production is shown in **Fig.11**. Due to the sharply increased population and GDP, the increased water shortage of agricultural water contributes to the decreased crop yield (**Fig.12**b), which also results in the stagnant farmer's income (**Fig.12**c). The increased water shortage happens because of the socioeconomy-ecology linkage, the increased ecological streamflow reduces crop water supply. The stagnant farmer's income happens because of the dual effect of both decreased crop yield and increased population. The total value of the primary industry is considerably related to crop yield. The reduced crop yield increases the food price, but its rate is still less than the rate of population growth. As food yield and income are greatly related to people's survival, the stagnant income and decreased crop yield will finally decrease carrying capacity and further intensify the overload index (**Fig.12**d). If the growth rate of the predicted population decreases (stage 3), there will be less pressure for water supply and can well balance the agricultural and streamflow water, further contributing to stable crop yield, increased farmer's income, and decreased overload index. Hence, how crop yield affects socioeconomy in this linkage can be embodied by the following three aspects: Firstly, the decreased crop production may lead to food crisis to come extent, which contributes to decreased population because of the limited access to food; Secondly, the main source of farmer's income is the total value of the primary industry, which is directly embodied by food yield, and less income caused by decreased food yield make it hard for farmer's survival; third, the declined population also decreases the labor force, which also hinders the socioeconomic development.

So far, the linkage of socioeconomy-food, socioeconomy-ecology, and ecology-food were all presented, which indicated that the three components interact and respond with each other.

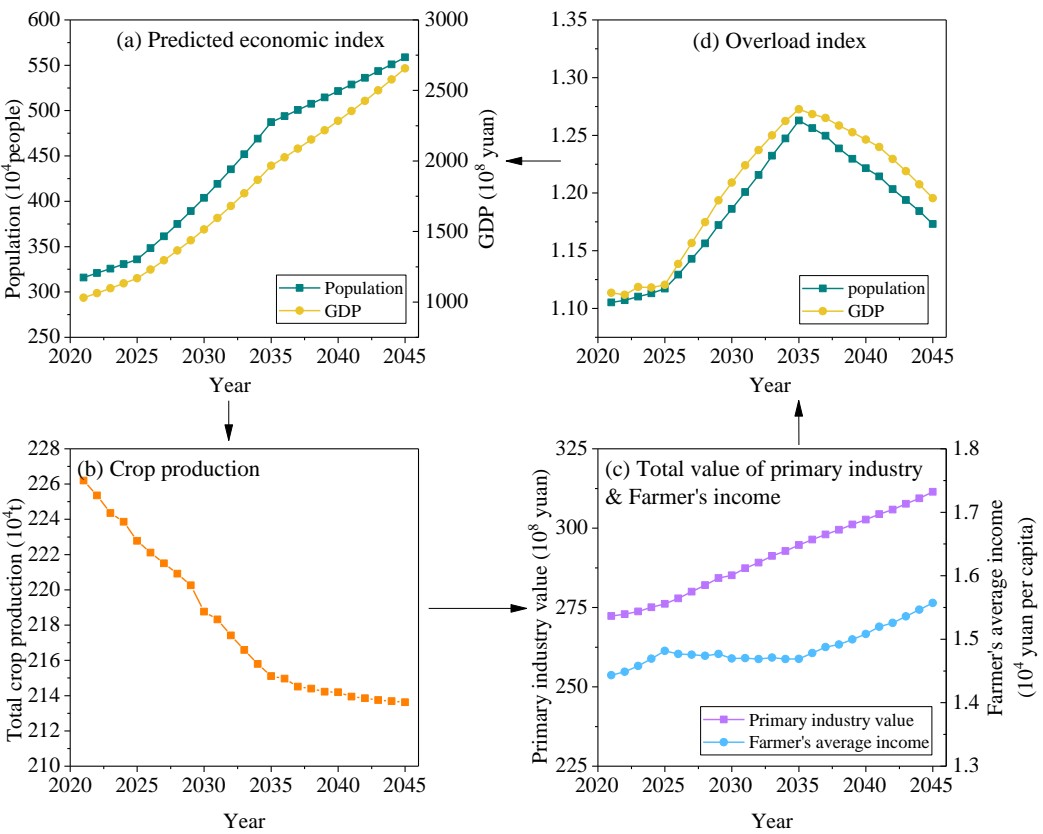

**Fig.12** Socioeconomy-food response linkage

*4.3 Assessment of coordinative degree of each subsystem and SDD*

The calculation result of SDD of SEF nexus and coordination degree of the socioeconomy (SOCEMY), ecology (ECLGY), and food (FOOD) is demonstrated in **Fig.13**. We can see that the variation of the four variables is also showing the state characteristics. The SOCEMY in the first stage is increasing, but it had an either decreasing (UGRB, Guilin urban area, Lingui, etc.) or stable (Xing'an, Yangshuo) trend in the second stage, indicating the coordinative

status of socio-economy is not good caused by the excessive growth rate of the economy. The decreased coordinative status of the socioeconomy subsystem also influences other subsystems and the SDD of total SEF nexus, reflected by the decrease of ECLGY, FOOD, and further SDD. Fortunately, the decreasing rate of ECLGY is smoother compared with that of FOOD, indicating the performance of the ecology of UGRB is relative well compared with socioeconomics and agriculture. This performance could be due to the dual effect of increasing streamflow water and

decreasing vegetation irrigation. The same was true for other administrative regions of UGRB. Moreover, for the whole basin, the value of SOCEMY in the later period of the second stage (about 2033~2035) is even lower than FOOD and ECLGY. From the perspective of administrative regions, it is more obvious in Guilin urban area, Pingle, and Lipu counties. It happens because the economic-stressed stage has lasted almost ten years in 2035, which is similar to the "pendulum model" that takes the effect that the pendulum "swings" towards the economic-stressed


system (See 2.1). As socio-economic index increases sharply and continuously, the ecological protection mechanism
will also be continuously triggered to increase the overload index, resulting in both SOCEMY and SDD reached the
minimum.

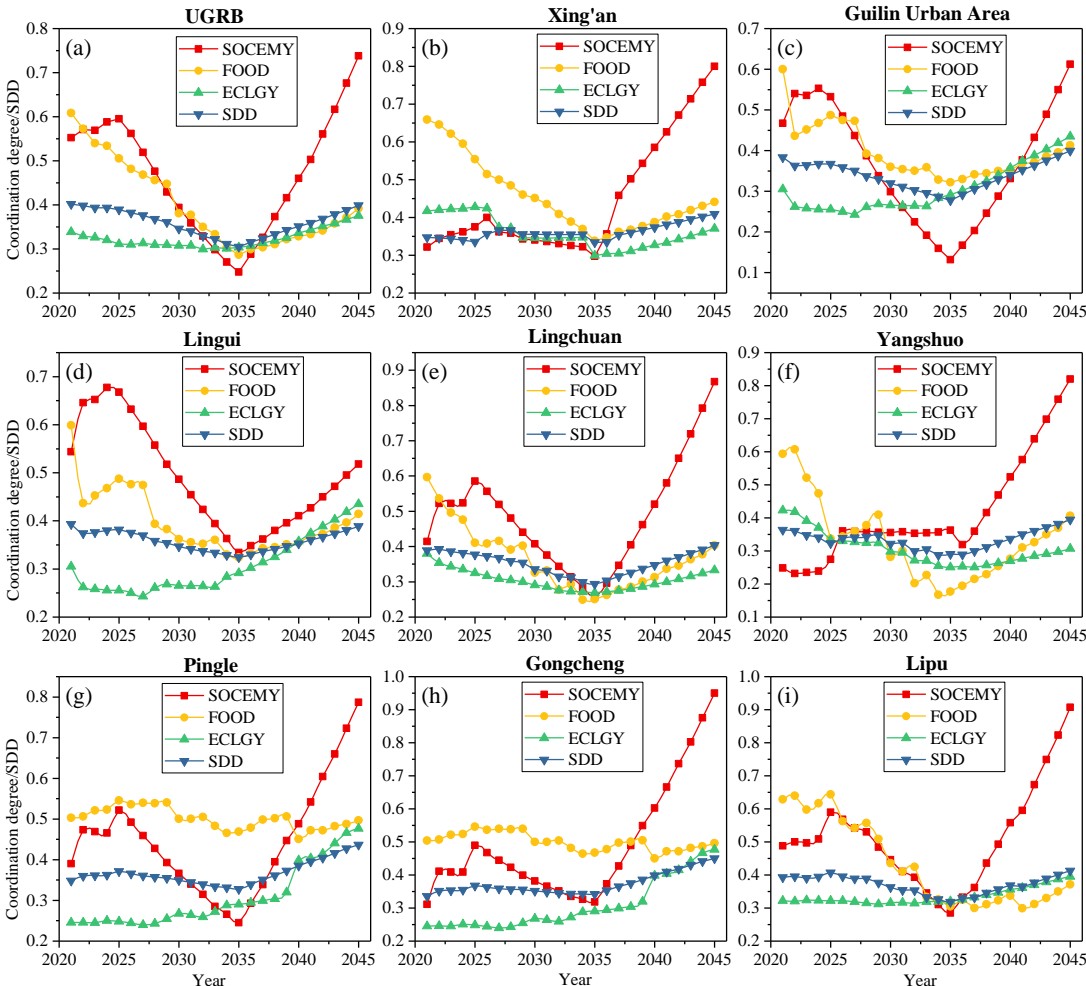

**Fig.13** Time variation of sustainable development degree (SDD) of SEF nexus and coordination degree of each agent

When it comes to the third stage, the value of SOCEMY increases, indicating the coordination of the
socioeconomic subsystem is improving. It revealed the decreasing of overload index and the increased carrying
capacity due to the relatively slower increasing rate of water demand of economic agent. The increasing value of
SOCEMY even promotes the coordinative degree of ecology and food, and the value of SDD is consequently
increased, revealing that stable economic growth will promote the sustainable development of SEF nexus. The good

phenomenon of the last stage happens because the relatively slow growth rate of water demand for the economic
agent will generate more water for food and ecology, and the increasing sewage and recycled water treatment rate
will provide relatively more water for users. The coevolution process assumes the "pendulum model" presented by
Van et al. (2014) and Kandasamy et al. (2014), where environmental awareness has been raised, and a stable



population rate occurred in the last era. The result presented in this study is similar to the findings in Van et al. (2014) and Kandasamy et al. (2014). Furthermore, we can speculate that in the 2045s, the pendulum of ULRB will also "swung" back to the stage of protective resources & environment and stable development of socio-economy.

## 5. Discussions

### 5.1 The reasons for coevolution trends and model performances

The overall coevolution changes and performances are affected by the external drivers embodied by the growth rate of population and GDP (see **Table 3**). The sharply increased rate in the second stage exactly corresponds to the era that "heavy government policy support and investment" and "population grow rapidly", which stressed in the "pendulum model" by Kandasamy et al. (2014) (see Section 2 and **Supplementary material S1**). The growth rate from 2036 to 2045 is lower compared with that from 2026 to 2035, which corresponding to the era of "remediation and emergence of the environmental customer". That's why the coevolution process of all the items depicts the characteristics of different stages. Although the optimal model is used in this study, the objective function of both socioeconomic and food agents still increases in stage 2, accompanied by the decreased food yield (**Fig.12**b), increased overload index (**Fig.9**f), and even lower SDD (**Fig.13**). This is because the optimal model is just the crucial tool for achieving sustainable water use in which the ecological agent is an indispensable part. Ecological streamflow must be guaranteed to maintain river health and, hence we can see the river streamflow increases rapidly in this stage simulated by the optimal model, and further intensifies the water shortage.

The positive feedback loop in the SD model (see Fig.4) will take effect if the optimal model is not coupled with the entire framework. However, this positive linkage will lead to divergence in the socioeconomic agent subsystem. That is, both water supply and population will increase circularly, and likely to result in unlimited growth of socioeconomy, which directly reduces the river streamflow and cause severe ecological problems. The socioeconomy/food agent and ecological agent constitute the negative feedback loop, and the optimal model will then be coupled with SD to help find the balance from this loop. With sharply increased population/GDP, the optimal model intensified the ecological streamflow to ensure river health. The optimal model does help to try its utmost to achieve the sustainable goal but there is no guarantee that the ideal status (higher SDD) will be achieved. The accelerated growth of water demand, caused by a rapid growth rate of population, is the main factor for the negative performance of the SEF nexus (e.g., high overload index). Fortunately, the situation in stage 3 has been improved, with decreased overload index, stable streamflow, and food yield. The moderate growth rate contributes to more water supply for supporting reasonable economic development. We can conclude that the technologies (such as optimal approaches) are just a tool that helps water sustainability, but management regimes and policy adjustment on external drivers is the fundamental approach to achieve this goal.

### 5.2 Decision making performance considering model uncertainty

The chain of the model is complex and usually contains lots of uncertainties, and decision-makers usually aim to achieve multiple performance objectives and have to make tradeoffs among those conflicting objectives, which arises from uncertainties (Herman et al., 2014, 2015). For uncertainties of multi-objective model, it is reflected mathematically by the portfolios of all the non-dominant optimal solutions (also called Pareto frontier) (**Fig.14**). Each dot in **Fig.14**, correspond to a certain weight vector $r=(\alpha_1,\alpha_2,\alpha_3,\theta)$, represents one possible alternative. Therefore, how to choose those optimal solutions from the Pareto alternative is the main source of the model uncertainty by



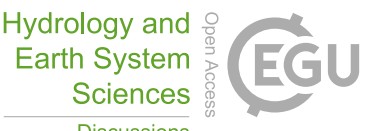

which the weight of each objective is reflected (Tingstad et al., 2014; Liu et al., 2019), that is, the tradeoff analysis. This study provides several alternatives based on different weighting factors to assess model performances. Twelve alternatives are presented in

Table 4 and represent the preferences of decision-makers, and the different performances are shown in Fig.15.

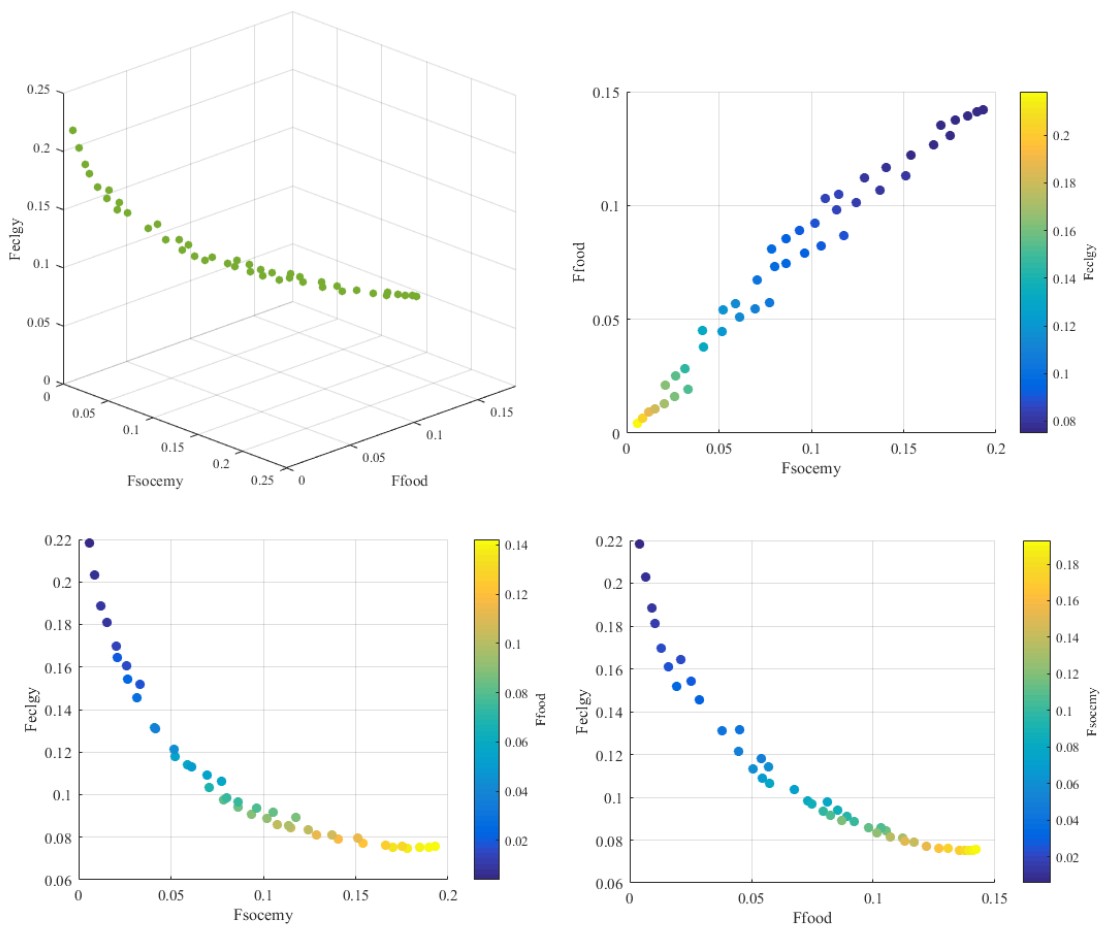

**Fig.14**   Portfolios of all the non-dominant optimal solutions of 2020.

Table 4   Twelve alternatives based on weighting factors for uncertainty assessment

| Alternatives | Weighting factors | | | | Alternatives | Weighting factors | | | |
|---|---|---|---|---|---|---|---|---|---|
| | $\alpha_1$ | $\alpha_2$ | $\alpha_3$ | $\theta$ | | $\alpha_1$ | $\alpha_2$ | $\alpha_3$ | $\theta$ |
| A1 | 0.2 | 0.1 | 0.2 | 0.5 | A7 | 0.2 | 0.2 | 0.4 | 0.2 |
| A2 | 0.2 | 0.1 | 0.3 | 0.4 | A8 | 0.5 | 0.2 | 0.1 | 0.2 |
| A3 | 0.2 | 0.2 | 0.2 | 0.4 | A9 | 0.4 | 0.2 | 0.2 | 0.2 |
| A4 | 0.1 | 0.2 | 0.4 | 0.3 | A10 | 0.5 | 0.1 | 0.2 | 0.2 |
| A5 | 0.2 | 0.1 | 0.4 | 0.3 | A11 | 0.4 | 0.1 | 0.2 | 0.3 |





| A6 | 0.3 | 0.1 | 0.4 | 0.2 | A12 | 0.25 | 0.25 | 0.25 | 0.25 |
|----|-----|-----|-----|-----|-----|------|------|------|------|

Approximately, A1 to A3 focus more on ecological streamflow with higher θ, while that of A4~A7 and A8~A10 is lower. A4~A7 focus more on food agent while A8~A11 focus more on the economic agent. A11 focuses on both economic and streamflow issues. A12 is the average level that each weight is set as equal. The value of both objective functions of each agent and SDD under each alternative is shown in Fig.15. From Fig.15, we can see that the values of SDD under A1~A5 and A11 are smaller than those under other alternatives. Meanwhile, the objective function of

both economy and food agents under A1~A5 and A11 is higher than that under other alternatives, suggesting more water shortage. On the contrary, the objective function of the ecology agent shows the opposite trend. We can contribute this result to the relatively higher weighting factor of θ and the lower weighting factor of α in those alternatives, resulting in the relatively less water serving for economic and food agents. Moreover, of all the alternatives, A12 performs the best with an equal value of weighting factor (0.25), suggesting that equal consideration to each agent is more likely to attain sustainable development. The value in other alternatives is either more or less

than 0.25, suggesting that excessive or lower weighting factors prevent the sustainable development of water resources to some extent. Therefore, the uncertainty analysis can also give a strong reference for the decision-making process for water resources management.

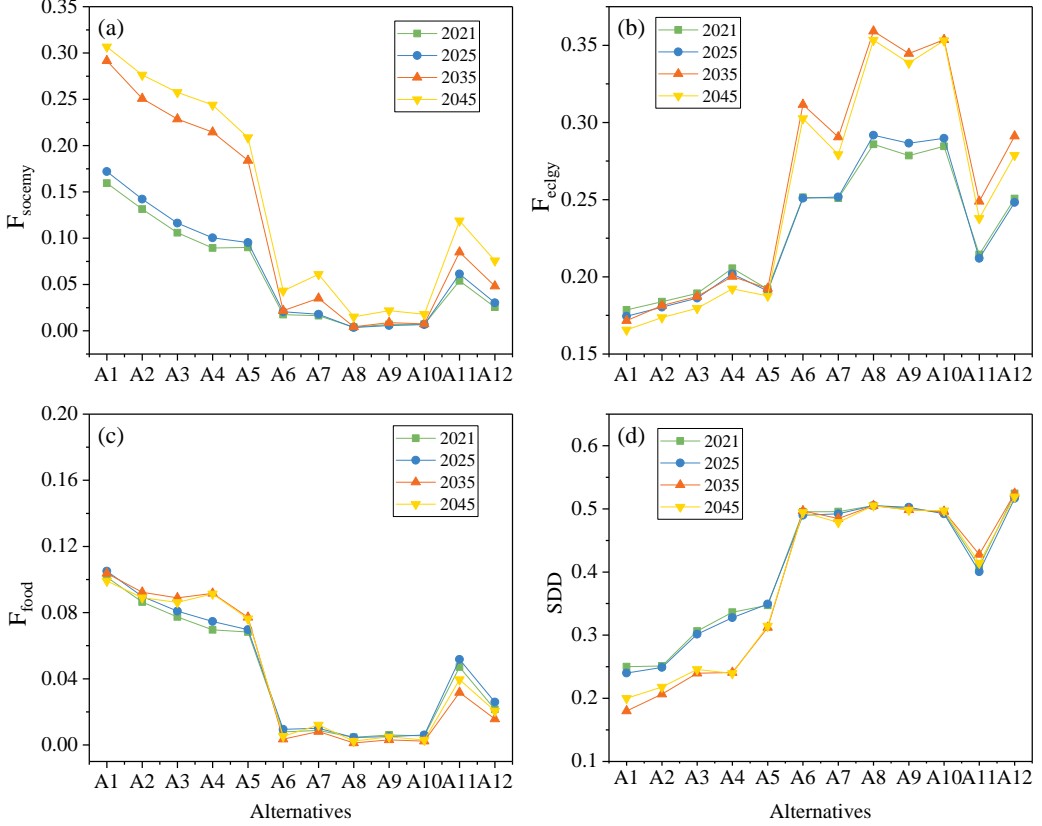

Fig.15  Sustainable development degree of different alternatives





### 5.3 Robustness analysis for SEF nexus

The key factor(s) that affect the robustness of the SEF nexus system is/are assessed to improve its reliability. The alternatives of A5, A7, A9, A11 are set particularly by controlling relative variables to assess the robustness of SEF nexus. In the case of both A5 vs. A7 and A9 vs. A11, we change $\theta$ while $\alpha_1$ and $\alpha_3$ remain unchanged to assess the influences of river ecology water changes on the performance of the SEF nexus. While in the case of both A5 vs. A11 and A7 vs. A9, we change $\alpha_1$ and $\alpha_3$ while $\theta$ and $\alpha_2$ remain unchanged to assess the influences of water changes of both economic and food agents on the performance of SEF nexus. According to Fig.15, the differences between both cases are shown in Table 5. To illustrate, the SDD value of 0.06 in row "A5 vs. A11" and column "2016" means that the difference of SDD value between A5 and A11 in 2016 is 0.06. From Table 5, we can see that the values in the lower two rows are smaller than those in the upper two rows. It indicates that when the weighting factors of both socioeconomic and food agents are certain, changing the weighting factor of streamflow will have a relatively significant impact on the performance of the SEF nexus in both objective function and sustainable development degree. Additionally, changing the weighting factor of both socioeconomic and food water uses will have less influence on model performance. In other words, the streamflow agent has a relatively great influence on the robustness of the SEF nexus model.

Table 5   Comparison of the performance of SEF nexus between different alternatives

| Case comparisons | Uses | $F_{socemy}$ | | | | $F_{food}$ | | | | SDD | | | |
|---|---|---|---|---|---|---|---|---|---|---|---|---|---|
| | | 2016 | 2020 | 2030 | 2040 | 2016 | 2020 | 2030 | 2040 | 2016 | 2020 | 2030 | 2040 |
| A5 vs. A7 | Influence of changing river | 0.07 | 0.08 | 0.15 | 0.15 | 0.06 | 0.06 | 0.07 | 0.06 | 0.15 | 0.14 | 0.17 | 0.16 |
| A9 vs. A11 | ecology on SEF performance | 0.05 | 0.06 | 0.11 | 0.12 | 0.04 | 0.05 | 0.05 | 0.05 | 0.09 | 0.10 | 0.12 | 0.12 |
| A5 vs. A11 | Influence of changing | 0.04 | 0.03 | 0.08 | 0.06 | 0.02 | 0.02 | 0.02 | 0.02 | 0.06 | 0.05 | 0.07 | 0.07 |
| A7 vs. A9 | socioeconomy and food on SEF performance | 0.01 | 0.01 | 0.03 | 0.04 | 0.00 | 0.01 | 0.00 | 0.01 | 0.01 | 0.01 | 0.01 | 0.02 |

The robustness of river ecology can also be reflected in the model performance of different years. From Fig.15, we can also see that both objective functions and SDD under A1~A5 have a greater difference between 2021&2025 and 2035&2045 compared with other alternatives. There will be a rapid economic increase from 2025 to 20305 and the ecological awareness in these alternatives outweighs other alternatives (with higher $\theta$), which is more likely to trigger the adaptive adjustment of the complex system and further accelerates the river streamflow. Then, there will be not enough economic water services, and the overload index will increase, further decreasing SDD in 2035 compared with 2025.

### 5.4 Simplifications of model dynamics and limitations

The proposed model simulates the dynamic evolution and feedback loops based on the three agents: socioeconomy, food, and ecology. The result proposed in this study is quite similar to Kandasamy et al. (2014) because he stressed that environmental awareness arises when an accelerated population is about to consume freshwater and results in the decrease of the population to protect the environment. This study also proposed the stable status of sustainability of both social productivities and environmental issues because the population growth rate is moderate and steady in the third stage to pay more attention to environmental awareness.

These individual three items are also prominent aspects or disciplines that contain numerous basic principles. Therefore, several assumptions and simplifications are often conducted to develop the nexus models that are, to some extent, one of the most necessary and significant ways for natural resources management practices for sustainable



development. For example, crop yield and primary industry that belong to agricultural productivities are determined by the original external conditions: precipitation and potential evapotranspiration that belongs to climate. The long-term historical climate data is used for the input of the model based on the assumption that the long enough historical data (monthly or even daily, several decades) can represent all possible climate scenarios. Not only SEF nexus but also other nexus is also based on several assumptions that are not always perfect. For example, the WPE (water-power-environment) nexus developed by Feng et al. (2019) considered water use quotas based on the exponential assumption. Still, they ignored the dependence on population growth rates.

The above two examples are purposed to illustrate that several assumptions should be often conducted before developing a certain model, which is also a key procedure of most scientific researches. But some assumptions may ignore some of the basic principles and further limit the models (Pindyck, 2015). For example, for the ecological aspect, it not only includes vegetation and streamflow but also consists of the water ecology and the issue that the human activities may lead to the ecological damages (Factories may increase $CO_2$ emissions), as well as the pollution from agricultural productivities. The population and GDP growth, to which the water demand of economic agents is related, is calculated by the Malthusian model (**see Supplementary material S2**) that may work only on a short time scale (about 2~3 decades). Still, it may not hold for a long time (100 years for example) scale. Those are all the model limitations that can be considered in our future research since the current study has simplified some of the basic principles as noted above. Therefore, no model is absolutely accurate and perfect, and several assumptions should be considered, although those assumptions sometimes are not that adequate (Pindyck, 2015). However, assumptions are the necessary procedures in most scientific researches. Thus, modeling approaches must be developed to solve a certain problem. Despite the limitation stated before, it does not mean that it is of no use or to give up entirely on estimating the sustainable development status more generally (Pindyck, 2015). We need to take advantage of the positive effect of a certain model to solve a certain problem.

## 6. Conclusions

This paper presented a new integrated framework that is used to analyze the dynamic interactions within coupled human and natural systems in the context of socio-economic development, food safety, and environmental protection by establishing system dynamic and optimal modeling. The system dynamics gives how the dynamic status of water supply performed, while optimal modeling gives insights on how the sustainable water uses can be achieved. The dynamic optimal results are generated by inputting the initial result of SD model of each time step and iteration process. The changing external conditions, i.e., the socio-economic development changes, result in nonlinear and multiscale feedback responses. The uncertainty analysis is also helpful for multiple tradeoffs and robustness analysis in the decision-making process. The result can give a firm reference and provide a practical tool for sustainable water use from the following two aspects:

This coupled modeling tool enables the coevolution and feedback by generating the whole scale of future trajectories that reveals the interactions across socioeconomic development, food safety, and ecological protection in a dynamic and optimal way. All the trajectories differed in different stages. That is, depending on the external drivers in terms of different stages, the dynamic changes manifest differently in water supply, streamflow water, farmer's profit, and population size. There are no obvious changes in the performances of the model in the first stage. In stage 2 (2026~2035), the severe increase of economy intensifies the interaction of a complex system by triggering the more streamflow water of reservoirs for the ecological agent. It results in less water for agriculture and social economy and cannot afford the rapidly increasing population and economy (increased overload index), and decreased food



yield. In stage 3 (2035~2045), with respect to moderate development of socio-economy, the interaction of the nexus
system will be alleviated, that is, the changes of streamflow water will tend to be stable, and there will be more water
to support the proper population size and economy, as well as crop yield. In terms of sustainable development degree,
the increasing trend occurred on stage 3 compared with the declining trend in stage 2. These results suggest that only
considering the economic benefits (stage 2) will rather accelerate the overload process of the overall system, which
inversely affects the socio-economic development and cannot achieve sustainable water use. If ecological awareness
arises and the economic growth rate tends to stable, it will be beneficial for the sustainability of water. Thus, the
coevolution process and dynamic interactions between human society and natural systems can provide valuable
information and guideline for policymakers on how to decide the development degree and manage water resources
on a regional scale considering economic development, food safety, and ecological protection.

The uncertainty analysis result of the coupled model also revealed the different performances considering the
need of various stakeholders, giving references to multiple tradeoffs influencing integrated systems and stakeholders,
notably the tradeoffs between water for social development, food production, and ecological protection. The Pareto
portfolio of the multi-optimization model based on different weighting factors reveals the competitive mechanism of
the three agents of the coupled model. The alternatives based on different weighting factors show the varied
sustainable development degrees and objective functions of each agent. Of all the alternatives, the equal consideration
of each stakeholder (weighting factor) is more likely to achieve sustainable development (with the greater SDD).
Therefore, policymakers can explore the future water allocation scheme among different needs of stakeholders based
on those different alternatives. Of all the agents within the integrated system, the river ecological part is more likely
to influence its robustness. This result suggests that the ecological agent of the integrated water resources system
should be paid more attention to the process of both water allocation and the policymaking process. The integrated
modeling framework presented in this paper is designed to simulate the interactions and feedback responses across
multiple agents, and the uncertainty analysis can improve the model reliability to provide valuable predictive insights
into the decision-making process of integrated systems.

**Acknowledgements**: The project was financially supported by National Key Research and Development Program of
China (No. 2018YFC1508200), National Science Foundation of Jiangsu (No. BK20181059) and China Scholarship
Council. The authors were also grateful to the sources of hydrological and meteorological data from hydrological
authority and statistical bureau, and the organizations and comments handled by Dr. Zengchuan Dong and Dr. Sandra
M. Guzman. The authors are still grateful to the insights and views of the editors and reviewers.

## Supplement: Supplementary materials (Data availability)

(Supplementary materials, uploaded to the supplementary links of journal's website)

## Author contribution

Yaogeng Tan prepared the manuscript and developed the model. Zengchuan Dong revised the manuscript.
Xinkui Wang and Wei Yan helped collect the data.

## Competing interests

The authors declare that they have no conflict of interest.



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
