# Peer review of "Identifying the dynamic evolution and feedback process of water resources nexus system considering socioeconomic development, ecological protection, and food security: A practical tool for sustainable water uses"

_Hydrology and Earth System Sciences, 2021_

## Referee Comment (RC1)

[referee-annotated manuscript omitted]

---

## Author Response (AR1)

**Author's responses to referees**

Note: Initial responses are in red; New responses (after revision) are in blue.

**Reply to referee #1:**

We are greatly impressed by the reviewer's decent tracks to our paper and give an opportunity to revise the paper. Below please find the point-to-point responses to your comments and help us to improve the quality of the paper.

1. There is a little bit of a poor connection between the first two paragraphs. The first paragraph discloses the background of the paper while the second paragraph outlines the main methods to solve the problem of water resources systems. From the perspective of the entire content of both paragraphs, the first paragraph implies the complexity of the water resources system that is reflected by multiple water uses while the second paragraph implies how to solve this problem. But the first sentence of the second paragraph (L48-50) is a little bit confusing and should tell the readers why "optimal algorithms", "decision support system", "MCDA" (or SAA) are suitable for tackling "water resource problems facing multiple water users". The general sentence that describes the characteristics of the water resources system (e.g. multiple water uses) is needed at the beginning of the second paragraph to connect the logic and then "SAA" is used to solve the problem. The details are marked in the preprint that is uploaded in the supplement.

Response: I totally agree with your opinion about your suggestions for the first two paragraphs. As the referee mentioned, the first paragraph disclosed the research background and research question, and the following paragraph should be the main methods that overcome this research question. The systematic analysis approach is exactly the main method to solve the systematic problem, exactly for water resources systems facing multiple objectives with high dimensions.

We have modified the statement as your requirements, see the revised version. (Tracked version: L59-71; Plain version: L53-58)

2. One or two sentences that further describes the optimal approach should be added in Line 59 to better connect the former (complexity of water resources system) and latter (disadvantages of optimal approach) information. I do know the optimal approach is the most effective approach to deal with the water management problems, but from the perspective of English language expression and rhetoric, the combination of optimal approach and water resources system should be added in Line 59 because the title includes "water resources system". Actually, as the author rightly mentioned in Line 40-41, water resources system "is susceptible to the influence of external conditions". From the system perspective, external conditions are influencing the status and performance of a certain system and is able to stimulate both the entire system and its each agent to adjust and strengthen themselves to better adapt to the external changes, which can be explained by "complex adaptive system" theory (See Holland, 1995). Optimal approach can describe and quantify such an adaptive process.

Response: We agree with your suggestion, and will add the relative sentences that describe the

characteristics of the optimal approach. I have also read the reference you mentioned (Holland, 1995) and optimal approach is essentially the adaptive adjustment of a certain system that can be summarized as a "complex adaptive system".

In this revised version, we have added the following statement in the text, see line 61-64 (plain) or line 75-78 (tracked).
"In addition, the optimal approach is essentially the adaptive adjustment of a certain system, which can be summarized as a "complex adaptive system" (CAS) (Holland, 1995) that is susceptible to external conditions. As for the water resources system, external conditions are able to stimulate both the entire system and its agents (i.e., water users) to adjust and strengthen themselves to better adapt to the external changes."

3. L67: There is lack of characteristics of nexus system, and should be described in two or three sentences. For example, its classification, their components, and feedback mechanism, etc. See Zhang et al.

Response: We have thoroughly read the reference from Zheng et al. Actually, the definition of nexus thinking can be classified into two categories: First, the nexus is interpreted as the interactions among different subsystems (or sectors) within the nexus system. Second, it is presented as an analytical approach to quantify the links between the nexus nodes.
The feedback mechanism not only includes the inner features of the coupled system by capturing the interactions between different sectors but also the external forces or actors that drive nexus system dynamics.

We have added the following text in L87-92 (tracked) or L73-78 (plain):
"The definition of nexus thinking can be classified into two categories (Zhang et al., 2018): First, the nexus is interpreted as the interactions among different subsystems (or sectors) within the nexus system. Second, it is presented as an analytical approach to quantify the links between the nexus nodes. The feedback mechanism not only includes the inner features of the coupled system by capturing the interactions between different sectors but also the external forces or actors that drive nexus system dynamics."

4. The novelty and characteristics are not purely extracted and the relevant language should be more concisely. For example, in line 100-104, the authors should describe the effects and advantages of the method that couples SD and optimal model, instead of just saying "identify the coevolution process and dynamic interactions". Actually, optimal model can give a water allocation scheme in an optimal way but cannot simulate the dynamic process precisely. If coupling it with SD, accurate coordination among different water users can be realized under the dynamic interactions of water resource nexus system, and further improves its reliability.

Response: Thank you. It is the key point. We agree with your opinion and will add such opinions in my paper (at the end of the Introduction section).
I have added your opinion in my revised paper. It is in line 117-118 (tracked version) and 101-103 (plain version). Also, I have rewritten the last paragraph of this section to better extract the paper's

novelty and characteristic.

5. Section 2.1 is the crucial part of the entire paper and should tell the readers why coupling SD and optimal models are applicable. It is necessary to start from the expression of the interaction of external changes on the nexus system and the advantages (see point 4) of both models to describe the reason why coupling SD and optimal models are needed.

Response: This is exactly the most crucial part of this paper. As our paper mentioned, the external drivers can be summarized by the pendulum model. The water resources system is exposed to external drivers and will influence the status of the system. The system dynamic model is exactly the powerful tool to simulate the dynamic interaction of the water resources system and its components. However, according to the theory of "complex adaptive system" (CAS), the external drivers not only influence the system's status but also starts the self-adjust process of both the whole system and its components to attain the adaptive status. Such a process can be characterized by the optimal model that can consider the coordination process among multiple agents, but it is unable to simulate the dynamic interactive process in a precise way. That's the reason why the SD and optimal model should be coupled.
The abovementioned information has been added in my revised paper as an individual paragraph in Section 2.1. See Line 145-154 (tracked) or Line 124-133 (plain).

6. In section 2.2.4, Line 220-223 and 229-232, it is not clear what the meaning of the symbol "+" and "-" is. I guess it represents increase and decrease. But in the following figures (Figs. 3 and 4) the "+" and "-" denotes the positive and negative feedback linkage. Are increase/decrease and positive/negative the same thing? If not, please note the meaning of "+" and "-" in lines 220-223 and 229-232, otherwise, it is confusing.

Response: Thank you for your suggestion, In Figs. 3 and 4 the symbol "+" and "-" is exactly the positive and negative feedback loop. But in Line 220-223 and 229-232, We want to express the increase and decrease of a certain variable. We will modify the "+" and "-" to "↑" and "↓" to express them precisely and make it clear.
Please see the revised version in Line 243-245, 252-255 (plain) or 267-270, 277-280 (tracked).

7. Fig 1 outlines the coupling framework of the SD and optimal model, but the internal relationship is not so clear from this figure. That is, the optimal model should be based on the certain interaction simulated by SD model. I guess it is the negative feedback loop that drives the optimal model (as the authors mentioned in Line 625-626), but it still needs to make clear in the Methodology section. Also, consider add it in the abstract if necessary. (Because for readers, they may read the abstract and methodology instead of results & discussion)

Response: Thanks for the decent suggestion. Yes, it is the negative feedback loop that drives the optimal model. I will make clear it also in the Abstract and Methodology section.
See (i) Line 23-24 (tracked) or 19-20 (plain) in Abstract section: "The multi-objective optimal model is used to quantify the negative feedback loops of the SD model by generating the optimal scheme of different water users.",

(ii) Line 139-140 (plain) or 163-164 (tracked) "SD model includes positive and negative feedback loops and the optimal model is used to quantify the negative feedback loop of SD.",
(iii) Marked part of Fig.1,
(iv) L289-291 (plain) or L314-316 (tracked): "Therefore, the optimization model is presented in this study to reveal the negative feedback loop and then achieve the sustainable water uses of each agent (see next section) by inputting the initial simulated result of SD and iteration (Li et al., 2018)."

8. In conclusion, some expressions of the English language are not so precise, and please see my marked versions in the supplement.

Response: Thanks. I will revise it based on your tracked changes.
Please see all the track changes in Conclusion part. All the changes you mentioned in the comment have been done.

Technical comments:

1. Unify the terms of the entire paper. For example "nexus system", "complex system" (L735, L739). Are they the same thing? If yes, please unify them. Another case "food safety" and "food security"; "crop yield", "crop production", "food production". Different terms make the paper ambiguous, also they are (or exactly are) representing the same thing. (See my comment in Line 20)

Response: Yes. Thank you for your suggestion! We will check them to unify the terms.
The entire paper to express the complex system has been unified to "WSEF nexus system", and other different terms have been unified to "food security" and "crop yield", respectively.

2. Title: This paper is about the dynamic process and feedback linkage of a nexus system. I suggest the title should include that. See the marked version in the supplement.

Response: Good point! We will make changes.
The title have been modified as "Identifying the dynamic evolution and feedback process of water resources nexus system considering socioeconomic development, ecological protection, and food security: A practical tool for sustainable water uses".

3. Fig. 11: The x-axis should be from 2021 to 2045?

Response: I will make changes in the revised paper.
Done.

4. Samely Table 5: Why the year in this table is 2016, 2020, 2030, 2040? Typing error or what? It is not consistent with other figures or tables.

Response: It should be 2021, 2025, 2035, 2045. Sorry, typing errors.
Done.

5: Line 500: should be sections 5.2 and 5.3?

Response: I will make changes in the revised paper.
Done. See L533 or L594-595.

6. Other technical comments are list in the supplement.

Many thanks for your decent suggestions and tracks to our paper! We will thoroughly revise the paper to greatly improve the quality!
All done.

**Reply to referee #2:**

Thank you very much for your positive feedback to our paper and the comments posted last time (hess-2020-461). I have thoroughly revised the paper based on the last comments. And I believe the current comment can greatly help improve the quality of the paper. Here are the responses to your comments:

First, in section 2.1, the authors should add one or two sentences to explain the connections (or relationships) between external drivers (as authors stated as "pendulum model") and core methods (SD and optimal model). In other words, what are the mechanism and methodological bases of nexus changes driven by external changes? Please explain it in the revised paper, or it will be confusing.

Response: Thank you for this point. Actually, referee #1 also pointed out this issue, in point 5. The water resources system is exposed to external drivers and will influence the status of the system. The system dynamic model is exactly the powerful tool to simulate the dynamic interaction of the water resources system and its components. However, according to the theory of "complex adaptive system" (CAS), the external drivers not only influence the system's status but also starts the self-adjust process of both the whole system and its components to attain the adaptive status. Such a process can be characterized by the optimal model that can consider the coordination process among multiple agents, but it is unable to simulate the dynamic interactive process in a precise way. That's the reason why the SD and optimal model should be coupled. In the revised paper, we will make it clear.

In the revised version, we have added the following paragraph in Section 2.1. See L145-154 (tracked) or L124-133 (plain):

"The external changes, which are quantified by the abovementioned "pendulum model", are one of the main sources that affect the status of the entire WSEF nexus system. It not only influences the system's dynamic status but also starts the self-adjust process of both the whole system and its components to attain the adaptive status. For the former, the system dynamic (SD) model is exactly the powerful tool to simulate the dynamic interaction of the water resources system and its components. For the latter, the self-adjust process can be outlined by the theory of complex adaptive system (CAS) that is first addressed by Holland (1995). He stressed that CAS is developed based on the system theory, indicating that each agent has its learning ability and stress mechanism to the external changes, and then becomes a stronger agent through such self-adjust process, to adapt to the change of external environment. The self-adjust process of each agent is substantially the optimal process, and the system optimal approach is thereby the effective tool that can quantify such self-adjust process of each agent."

Second, as the authors stated in L88-90, "those methods are used to simulate the dynamic status and feedbacks just in an objective way but no optimal function inherently, which limits the goal of sustainable water uses to some extent". But in the following sections, I didn't see any qualitative or quantitative analysis and proofs about the advantages of the methods used in this paper compared with the current methods. Does it improve the model's reliability, or, achieve the coordination more accurately among different agents under external changes? Or either of the two models cannot

achieve the desired effect? Or other better effects? Such analysis should be implemented in the Discussion section to better enhance the contribution of the paper and better answer the research question.

Response: Thank you for your suggestion. In fact, SD model is used to simulate the dynamic status of a large system, and it can also reveal the dynamic interactions among the components under external drivers. However, only SD model is unable to ensure the coordination among each agent. That is, it does not ensure its best status, just a tool for simulating the dynamic changes. Then optimal model is used to attain the coordination status based on CAS theory. We will try to make quantitive analysis to verify this assumption in the discussion section. For example, comparing the value of some variable(s) (e.g. SDD, optimal function, etc.) under different condition (SD only, SD and optimal model).

We have added the corresponding analysis in the revised paper in section 5.1, i.e., the value of SDD under the different condition: a) SD only; b) SD & optimal model, and the performance of the latter condition is indeed better than the former. The results are shown in Fig.14. However, due to the time limit, the analysis of other variables (like optimal function) will be conducted in our further study, because it takes lots of time. (Tracked: L688-689, 695-699; Plain: L657-658, 664-667).

Third, some comparisons should be made between other studies. For example, I saw the relative research from Tan et al., (2019) which deals with a similar area using similar optimal approaches but, some results and conclusions are not consistent. For example, in that paper, the authors state that the socio-economic agent is more sensitive. But in this paper, they claim that river ecological agent is more likely to influence the model's robustness. Why do the different results happen? Please explain it. By the way, I didn't read the entire paper (Tan et al., 2019, Water, 11, 4) in-depth and just see its conclusion section. But I believe every reader will have this question if only read the conclusion part of both papers. They may not read the entire paper but the abstract and conclusion.

Response: A very good question. In fact, this is the different method of robustness analysis. In this paper, the robust analysis is based on the changes of the weighting factors. Many previous studies also used this method. For example, Feng (2019) established the integrated framework of water resources system and applied in Danjiangkou Reservior by introducing many parameters. The robust analysis is conducted based on the changes of these parameters, and the model performance (revealed by certain variables) under different value of these parameters are analyzed. In our study, the parameters are the weighting factors of the entire optimal model. But in that study (Tan et al., 2019), the robust analysis is conducted by changing the reservoir's streamflow and comparing the value of objective function of both in-stream and off-stream water users. The increasing streamflow results in decreasing water supply of off-stream, which lead to the higher increasing rate of off-stream objective function. In fact, they are like apples and oranges due to the quite different methods. Therefore, there are lots of ways for robustness and their core content is quite different, which leads to different results. In terms of robust analysis, both two studies just attempt to make the initial analysis, and in our further research activities, we can find more advanced methods of robust analysis along with more extensive literature review.

We have added such statement in the revised paper. See L727-738 (plain) or L759-770 (tracked).

**Reply to referee #3:**

Thank you for your positive remarks to our paper. We will thoroughly revise the paper based on your comments. Below pleased find out the responses to you comments:

1. The naming scheme of the "SEF" nexus, which also noted socioeconomy, ecology, and food, has no core basis. After all, our subject (or academic discipline) is hydrology & water resources. I do know water resources is one of the key support of these three items but, from the perspective of "nexus" naming, water should be included in the nexus terminology (or jargon), that is, water-socioeconomy-ecology-food nexus (WSEF nexus). Or it will be confusing.

Response: A decent point. We will make changes in the revised paper.
All the changes have been done.

2. Section 2.4 (start with L399): evaluation index system seems to be widely used in other studies and not specific. I don't know the difference when the same evaluation system is used in other studies because it's also suitable in other studies. So, how can the indicator evaluation system represent the sustainable development of a nexus system? (I don't know if my understanding is right, just my personal view.)

Response: Thank you for this question. The core content is to evaluate the sustainable water uses of the nexus system. So, there should be a quantitive evaluation on how the sustainable water uses is like. Only qualitive estimation cannot reveal the coordinative degree (or how the best/good/worst status could be defined), so the evaluation system that evaluates the sustainable water uses of a system were adopted in this study. This is just a tool to evaluate how sustainable status is like.

3. The paper lacks the calibration and validation part. Conceptual models should be calibrated and validated before using and simulating in a real case study. Please add such analysis (even the result).

Response: Yes. Model calibration and validation cannot be omitted. We will add this part.
The added part is in Section 4.1 (Plain: L495-501; Tracked: L524-531) and supplementary material S7..

4. The nexus system used in this paper is a case study of the humid region of south China but lacks universality analysis, that is, is the model only suitable for the humid region or all-region? Are dry regions also suitable?

Response: A good question. This study is exactly a case study of Southern China in order to verify the reliability and availability of the nexus framework. About the availability of other regions, we will try to do it in our future research. This is the main limitation of this study and we acknowledge it.
This is a key limitation of this study and have added in Section 5.4 (Tracked: L790-791; Plain: L758-759)

5. Results: Section 4.2.3: socioeconomy-food response linkage. It seems that carrying population/GDP is in direct proportion to crop yield from this paper. But in real cases, the relationship between crop production and carrying population is not as simple as a linear relation. Their relations are really complex and cannot simply be analyzed from a quantified trend. See Lyu et al., 2020.

Response: Thank you for this point. This paper mainly focuses on the dynamic interaction and feedback linkages of water resoures system. But in most cases, crop production is roughly in proportion to population sizes, because crop production supports people's survival. The socioeconomy-food linkage is presented based on this assumption. However, if this point is deeply explored, crop production and population size (or GDP) is not as simple as linear relationship, which can be another good research field and is beyond the scope of this paper. This point can also be the main limitation of this paper.
We have added it in Section 5.4 (Tracked: L782-785; Plain: L750-753)

6. L32: agricultural water uses have nothing to do with rainfall, it should be a "process of agricultural water demand". As authors rightly said in Section 2.2.3, food production is greatly related to Wp (Crop water demand, see Eq.9), which substantially related to ET0, instead of rainfall.

Response: We will make changes.
Done. (See plain version L36 and tracked version L42.)

7. L43: Add "However," before "the dynamic interactions......" to connect the logic. These two sentences have an adversative relation.

Response: We will make changes in the revised paper.
Done. (See plain version L47 and tracked version L53.)

**Reply to CC:**

The missing reference has been supplemented. See Line 1003-1004 (Tracked) and Line 968-969 (Plain).

---

## Author Response (AR2)

Thanks for both reviewers for the positive view for our revised manuscript, we have made all the changes to prepare for the final publication of this paper.

Below please find the point-to-point responses.

**Reply to RC#1:**

1. L102: "Therefore, coupling both systematic methods..." Which systematic methods? Please explain.
Reply: Modified as: "coupling systematic methods of both SD and optimization approaches can integrate ……" See L102-103.

2. L108: "optimal model" but some statements say "optimization model". Are they the same?
Reply: Changed to "optimization model" (L109), and "optimal model" in whole manuscript has been changed to "optimization model".

3. Fig3: What does the grey words mean?
It is "shadow variables". We have explained in L243-244.

**Reply to RC#2:**

1. Move the part in lines 727-738 (as point 3 in my first-round comment) to Section 5.4. It can be considered a research limitation.
We have moved to L748-759.

2. Remove other limitations in 5.4, and they are unnecessary: a) L753-757 about climate data and population growth rates. b) I don't understand what L760-L774 wants to imply. It seems that it was going a bit far. Please delete them.
Done.

Technical:

L668 remove "," before crop yield to express both streamflow and crop yield are stable. Otherwise, the readers do not know how the crop yield will be.
Done.

L807: WSEF nexus system, no "s".
Done. See L792.